environmental science/biogeography/ecology

connectivity, carnivore, environmental services, protected areas, landscape, Colombia

**Author for correspondence:**
Diego A. Zárrate Charry
e-mail: godiezcharry@gmail.com

# Connectivity conservation at the crossroads: protected areas versus payments for ecosystem services in conserving connectivity for Colombian carnivores

Diego A. Zárrate Charry[1,2,3], José F. González-Maya[2], Andrés Arias-Alzate[4], J. Sebastián Jiménez-Alvarado[2], Jessica Dayanh Reyes Arias[2], Dolors Armenteras[5] and Matthew G. Betts[1]

[1]Forest Biodiversity Research Network, Department of Forest Ecosystems and society, College of Forestry, Oregon State University, Corvallis, OR 97331, USA
[2]Proyecto de Conservación de Aguas y Tierras, ProCAT Colombia/Internacional, Calle 97a #10-67, Of. 202, Bogotá, Colombia
[3]Fondo Mundial para la Naturaleza WWF Colombia. Cra. 10a #69 A-44, Bogotá, Colombia
[4]Facultad de Ciencias y Biotecnología, Universidad CES. Cl. 10a #22-04, Medellín, Colombia
[5]Grupo de Ecología del Paisaje y Modelación de Ecosistemas ECOLMOD, Departamento de Biología, Facultad de Ciencias, Universidad Nacional de Colombia, Bogotá, Colombia

DAZC, 0000-0002-2794-4015; JEG-M, 0000-0002-8942-5157; AA-A, 0000-0001-9139-5690; DA, 0000-0003-0922-7298; MGB, 0000-0002-7100-2551

Protected areas (PAs) constitute one of the main tools for global landscape conservation. Recently, payments for environmental services (PES) have attracted interest from national and regional governments and are becoming one of the leading conservation policy instruments in tropical countries. However, the degree to which areas designated for PES overlap with areas that are critical for maintaining species' landscape connectivity is rarely evaluated. We estimated habitat distributions and connectivity for 16 of the 22 mammalian carnivores occurring in the Caribbean region of Colombia, and identified the overlap between existing PAs and areas identified as being important for connectivity for these species. We also evaluated the potential impact of creation of new PAs versus new PES areas on conserving connectivity for carnivores. Our results show that PAs cover only a minor percentage of the total area that is

important for maintaining connectivity ($x = 26.8\% \pm 20.2$ s.d.). On the other hand, PES, if implemented extensively, could contribute substantially to mammalian carnivores' connectivity ($x = 45.4\% \pm 12.8$ s.d.). However, in a more realistic scenario with limited conservation investment in which fewer areas are set aside, a strategy based on implementing new PAs seems superior to PES. We argue that prioritizing designation of new PAs will be the most efficient means through which to maintain connectivity.

# 1. Introduction

Protected areas (PA) and payments for environmental services (PES) are the two leading landscape conservation policy instruments used to decrease forest transformation and conserve biodiversity [1–4]. PAs do not typically use species distributions as main criteria for establishment [5,6]; instead, designation often applies coarse-resolution information on ecosystem and biome representation, which may be of limited use as biodiversity surrogates [7,8]. This approach has led to PA systems that focus primarily on untransformed natural ecosystems or biomes in places of reduced economic interest (e.g. high mountain ranges) that do not necessarily achieve biodiversity targets [8–10]. This lack of representativity has been identified at global and national scales [5–7]. Furthermore, policy instruments focused primarily on PES do not necessarily align with biodiversity or ecosystem distributions [11,12]; such a mismatch also reduces opportunities for fulfilling biodiversity conservation targets (e.g. Aichi Biodiversity Targets) [13].

PAs have been the primary biodiversity conservation strategy worldwide, with strong positive effects on forest and ecosystem conservation [14–16]. The total area of PAs has increased in tropical countries, from 12% in 1990 to 26.3% in 2015, an increase three times greater than in boreal and temperate zones [14]. On the other hand, in recent decades, PES has been presented as an economic incentive for developing countries to mitigate climate change by reducing deforestation and forest degradation, while protecting high biodiversity areas and improving local livelihoods [12,17]. The inclusion of poverty alleviation considerations within PES schemes and its environmental and economic potential impact could shift legislative and management efforts from PAs to PES, and may prioritize social poverty alleviation above environmental goals. Although it is not clear yet if PES emerging popularity and implementation is reducing the rate of PA establishment, there seems to be an increasing trend toward promoting PES conservation actions rather than pursuing strict protection [18,19], even though conservation success of PES may be reduced in terms of slowing deforestation rates at least [20].

Many biodiversity-rich countries still contain intact forests but are currently facing higher levels of deforestation, and therefore require urgent conservation focus [21]. Forest loss affects both area and landscape connectivity, thus impacting the persistence of species [22]. To maintain metapopulations, there is an increasing need to prioritize areas based on their suitability as habitat, landscape connectivity, long-term function and degree of fragmentation [23–26]. The degree to which landscapes are connected is dependent on individual species' biology [27] and the identification of such 'functional connectivity' entails understanding species-specific ecological requirements to identify appropriate types and arrangements of habitat patches selected for protection [28]. Additionally, given that PAs usually cover only a small percentage of the landscape, it is important to emphasize regional and national policy that focuses on the conservation of areas between reserves (i.e. the 'matrix') [29]. Species' dispersal, and therefore population viability, depends on movement among patches [30,31], which is constrained by inter-patch distances, matrix resistance and land cover quality [30,32,33]. However, historically, landscape planning has focused on the conservation of large patches in an attempt to mitigate potential effects of fragmentation and habitat loss [34].

Colombia contains some of the most endangered and unique ecosystems worldwide, and its tumultuous social and political history has facilitated the maintenance of essential forest areas with minimum anthropogenic impacts [35]. The main tool for protecting this biodiversity corresponds to the 314 052 km$^2$ of terrestrial PAs, which are distributed in 1323 areas within the Protected Areas National System (SINAP) which includes private and public, regional and national, PAs that, as of 2018, cover 15.16% of the country. Despite this considerable PA network, the country has identified eight potential new areas to be declared based on ecosystem and biome representativity and gaps in the existing PA network [36].

PES in Colombia are still not widespread; and even when the legal framework was already developed, clear protocols on how or where to implement PES are vague and diverse, and its execution depends on the funding source, implementation agency and regional factors [37–39]. The most extensive PES effort to date covers approximately 130 km$^2$; these areas are primarily located within small private farms (mean forest size = 0.15 km$^2$ [40]) that are scattered and opportunistically dispersed across the country. Existing PES

sites were not selected on the basis of either their location or size [37,38]. Location of selected farms is not systematic and is guided mainly by state-specific interests or by the voluntary commitment by landowners to the programme [37]. At the national and regional levels, there a is lack of harmonization between landscape conservation policy instruments such as biodiversity conservation targets and co-benefit opportunities [11,41]. None of the current conservation instruments consider the importance of landscape connectivity in selecting conservation areas [42]. Currently, the creation of new PAs in Colombia face considerable economic and social challenges, and the Colombian government is shifting efforts to alternative or complementary conservation schemes such as PES, a move which is strongly supported and driven by the international community [43].

In this research, we conducted a case-study spatial analysis to test whether existing and projected PAs and proposed PES schemes (i.e. water provision, carbon sequestration, risk reduction areas) are sufficient to protect habitat and its connectivity for a selected, ecologically important group of species. We focused our study on terrestrial mammalian species from the order Carnivora in the continental Caribbean region of Colombia (16 of 22 species). We selected this group for three reasons. First, habitat loss and fragmentation tend to have strongly detrimental effects on this taxon due the slow life history traits and greater habitat area requirements [44,45]. Second, these species are heavily impacted by conflict with humans, either through direct mortality, or via human hunting of their prey base [46,47]. Third, these species are officially recognized by institutions as vulnerable, have important cultural value to local communities in Colombia, and are therefore included in national and regional management plans [48,49].

Both PAs and PES schemes are currently represented in the region, but there is still discussion about the expansion of both; unfortunately, there are still no scientifically based evaluations on the relative efficacy of these approaches for biodiversity conservation.

# 2. Material and methods

We focused our analysis on the Caribbean region of Colombia, one of the five natural regions of the country [50]. The Caribbean region of Colombia is an appropriate study area for the identification of regional conservation needs given its exceedingly high biodiversity [15,51]. It is also representative of the rest of the country in that it is a data-poor region that is shifting from an underdeveloped to a developing economy [52]. The region extends from the Andes Mountain range in the south to Venezuela to the east and abuts the Caribbean Sea in the North and West. The region encompasses nine departments (political and administrative divisions analogous to states). Elevation ranges from 0 to 5568 m above sea level. The region area is 194 811 km$^2$ and includes four biogeographic provinces with distinct climatic, geomorphologic and ecosystem characteristics [50] (figure 1).

Our overall approach uses occurrence data, environmental and land-cover information and existing ecological knowledge, within a framework that reduces spatial bias in occurrence data [53]. This framework includes for each species: (i) quantifying potential distributions [54], (ii) identifying suitable habitat patches (based on national land cover cartography) [55], and (iii) defining a connectivity network between suitable habitat patches (based on ecological information), allowing a species-specific prioritization approach [56,57]. We then used the whole framework as an input for comparative landscape analysis to assess the potential effectiveness of PAs versus PES.

We developed this methodological framework to ensure a standardized approach for the identification of priority connectivity patches for multiple species in the context of data scarcity. This approach, and the resulting outputs, enables use by regional and local stakeholders within existing landscape planning instruments and cartographic information [58,59]. We used the same methods and followed the same previously published modelling and data management procedures for all species [54,57,60,61], thus allowing for sound inter-species comparisons.

## 2.1. Habitat identification

We defined habitat for each species in the region as the total area of all the suitable habitat patches that were (i) located within the modelled (climatically suitable) potential distribution of the species, (ii) continuous areas composed of land-cover types previously reported as habitat for the species, and (iii) patch sizes that were equal to, or larger than the reported home range for each species (figure 2). We choose this rule set to ensure that each identified suitable habitat patch had both the cover type, patch size and climatic variables to support each species (electronic supplementary material, S3) [62].

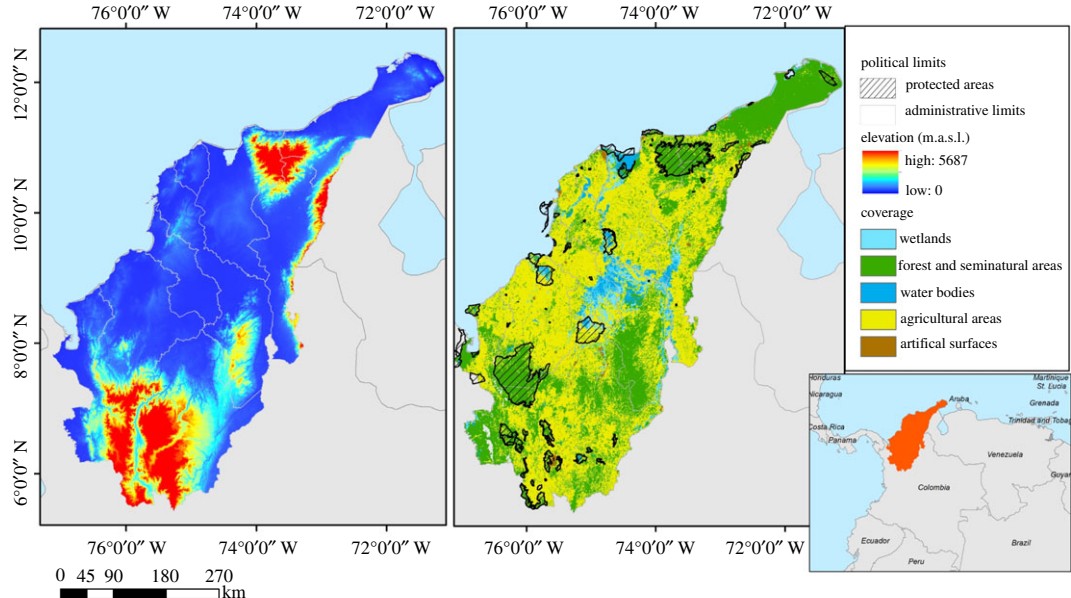

**Figure 1.** Map of the study area showing (*a*) the elevation, (*b*) the presence of protected areas, and cover classification.

We modelled only species with at least 20 spatially independent occurrences, which is already a small number of occurrences to fit a model [63]. Data scarcity in developing countries is pervasive [64,65], yet should not preclude preliminary assessments of species distributions. Even at this low threshold for inclusion, we had to remove three of the carnivore species from our analysis due to data scarcity. Since we lacked absence data, we could only use presence-background ecological niche models, which potentially have limitations regarding inference and accuracy [66].

To define the potential distribution of each species, we developed ecological niche models using the Maxent algorithm [67]. This approach uses a Bayesian model fitting procedure to predict the likelihood of the occurrence of a species' ecological niche as a function of the environmental variables that spatially coincide with where the species has been recorded [68]. The inherent sampling bias of presence-only data has been heavily criticized [69]. For instance, species records are more likely to occur near areas that have been more heavily travelled and surveyed. We therefore applied correction methods to reduce the effects of sampling bias, for each species.

To identify clusters of occurrences, we conducted species-specific spatial thinning based on the distribution of the existing occurrence records, used a pairwise distance matrix between occurrence points to identify potential data clustering [70], and created a species-specific background based on biogeographic units where the species has been reported [71]. We report the species-specific thinning used and the numbers of occurrences removed by the spatial thinning in the electronic supplementary material, S1. Before fitting models, we evaluated the correlation among environmental variables used to train models and removed those that were highly correlated (i.e. a Spearman rank correlation greater than 0.8). We validated the statistical significance of all variables used in each model using a jackknife analysis [67].

After selecting environmental variables, we estimated optimal model complexity for each species as follows. We ran 30 Maxent models per species changing the regularization multiplier and the combinations of the used feature classes. We used six regularization multipliers (i.e. 0.5, 1, 1.5, 2, 2.5, 3) and five feature class options (i.e. linear, linear/quadratic, linear/quadratic/product, linear/quadratic/product/threshold, linear/quadratic/product/threshold/hinge) [54,72]. The performance of each model was evaluated using a cross-validated masked checkboard subset of randomly selected occurrence points [54,60]. We used AIC to choose the best performing model for each species and retained only those with the area under the receiving operator curve (AUC) values higher than 0.7. We developed all models using the ENMeval package for R [54].

The final selected model was used to construct a binary output using the minimum training presence logistic threshold, which reduces omission errors [73]. We then refined this binary output using the maximum and minimum elevational limits for each species [49,51,74,75]. The 'potential distribution' or environmental niche is understood as the suitable abiotic environment of a species [76]. We include the detailed parameter and tuning information for each species' ecological niche model in the electronic supplementary material, S1.

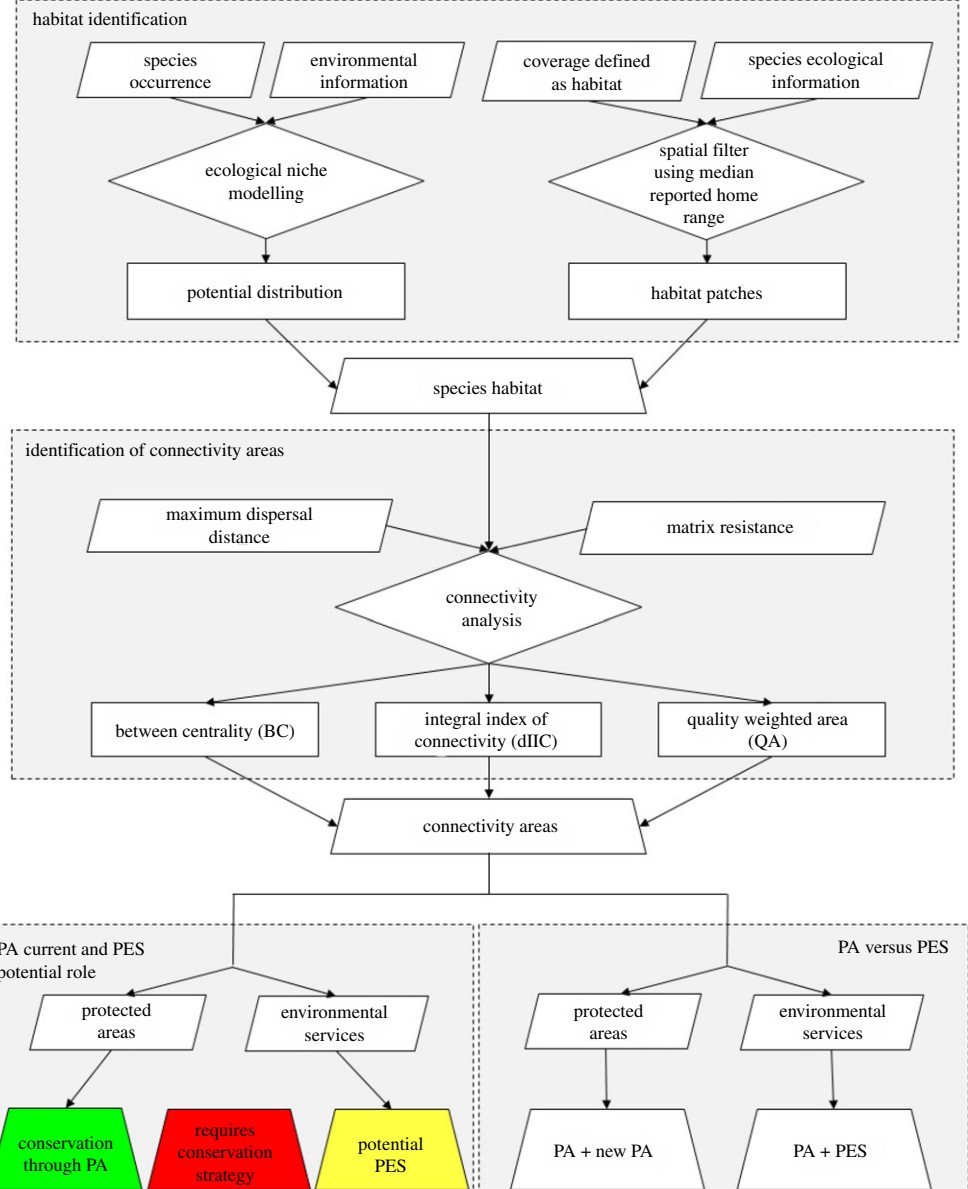

**Figure 2.** Methodological processes used to identify mammalian carnivores' habitat and connectivity areas and identify the current and potential conservation role of PAs and PES in the Caribbean region. The detailed explanation of each of the methodological processes is in the methods and results.

Within each identified potential distribution, we used regional land-cover data to identify all suitable habitat patches that had at least one of the cover types reported in the literature as being habitat for each species. We further filtered the final number of patches present in the region by removing those that were smaller than the individual median reported home-range of each species. We defined patches in this way to reflect the need for population connectivity (individuals must reside and survive in a patch during movement between subpopulations) [25,30]. We acknowledge that some species may also use smaller patches as temporary stepping stones [30], so our analysis should be considered conservative in this regard.

Although our results are based on the most up-to-date spatial information possible, the current land-cover data are still based on Landsat images from 2007. We therefore updated suitable habitat patches forest loss with up-to-date information on forest change between 2007 and 2016 [77].

## 2.2. Identification of connectivity areas

We identified connectivity areas by quantifying the 'habitat connectivity importance' of each patch for each species. The habitat connectivity importance refers to the role that each suitable habitat patch

plays in facilitating connectivity among all suitable patches [57]. We used a graph theoretical method to identify the connectivity importance for each suitable patch. Graph theory combines spatial habitat distribution data with information on the movement ecology of a species to identify areas that are potentially important for connectivity [56,57].

In graph theory, the landscape functions as a network of nodes (i.e. suitable habitat patches), which are connected by potential linear links; in our analysis, these linear links were calculated as the least-cost-path distances between patches, with lengths limited by the maximum dispersal distances of each species. To specify the least-cost-path distances, we used resistance layers developed from previously reported variables that are known to affect movement of terrestrial carnivores [78]. We also assumed that areas outside of a species' potential distribution impeded movement. We imposed a movement restriction for areas that fell outside of our quantitative definitions of 'habitat' for each species; for simplicity, we included habitat as a dichotomous rather than continuous (probability-scale) variable.

Together, resistance layers and non-habitat matrix were used to create a specific resistance matrix for each species (electronic supplementary material, S2 and S3). Some limitations may arise from the scarce information available for most species. We conducted an extensive review of the required habitat characteristics, but data are currently rare, and most datasets do not exist for Colombia. The only solution was using available data from other countries. We acknowledge that in some cases, local adaptations may result in regional differences in habitat associations and home range sizes [79]. Finally, we included all suitable habitat patches as nodes in the connectivity analysis. The graph analysis considered the connectivity of the whole network and weighted the importance of each node and link within it [80].

We used three indexes to quantify habitat connectivity importance, each of which reflects different attributes that are important for animal populations. First, between centrality (BC) identifies the role of a patch as a stepping stone based on the sum of pathways between all pairs of patches are connected to a focal patch [56]. Second, we used the node importance index (dIIC) to calculate the importance of each node within the whole landscape, and the potential decrease in connectivity when a particular node is removed [80,81]. dIIC uses as habitat connectivity the values from the integral index of connectivity (IIC). The IIC makes a binary connection model with unweighted links, providing landscape connectivity values that range from 0 to 1; a value of 1 represents a landscape composed only of habitat [82]. We calculated BC, IIC and dIIC using Graphab 2.0 [83]. Third, we used the quality-weighted area (QA), a value that represents an intrinsic patch attribute based on the amount of high-quality habitat within each patch. In this analysis, we identified the quality-weighted area (QA) as the patch quality (quality defined as the presence of natural or semi-natural coverage) multiplied by the size (in square kilometres) [84]. Prioritization of patches using different indexes can generate entirely different results. Furthermore, using several connectivity indexes can be redundant or allow interpretation of unique characteristics to each index sensitivity [56]. We selected and used these three indexes because they represent unique dimensions of existing connectivity indexes thereby avoiding redundancy [56].

We transformed each index value to ensure they have equal scales, and summed the values of the three indices to rank patches in order of connectivity. We selected patches in rank order of their connectivity importance score until we reached a total of 50% habitat conserved. This is the threshold established by Di Minin et al. [85] for effective conservation of carnivores. We followed the 50% value following the proposed adequate representation of species range protection presented by Di Minin and collaborators [85]. We acknowledge that this value seems arbitrary, but such threshold can be adjusted upwards or downwards to match available conservation resources, or other proposed conservation targets (e.g. 30% for the post-2020 Aichi conservation goals) [85,86].

## 2.3. Protected areas representativity and the potential role of payments for ecosystem services

We identified the degree to which the connectivity areas for each species, and all species combined, overlapped with existing PAs and areas that have been deemed important for PES. To identify current PAs, we used the World Database on Protected Areas v. 1.3 [87]. As areas of potential PES implementation, we used the environmental services spatial prioritization constructed by Rodríguez et al. [88], which uses the environmental services currently supported by the Colombian national legislation. Rodríguez et al. [88] identified areas within the range of one to four overlapping environmental services (water provision, regulation of water flows, landslide prevention, carbon storage). These areas do not currently have any legal protection, but they represent places were PES schemes could be implemented following the most recent national legislation.

For this analysis, we included any pixel with the presence of at least one environmental service as an area for potential implementation of PES. To evaluate the potential role of PES, we assumed that PES will

be implemented across a total area of 52 683 km$^2$, which represents 27% of the study region. We acknowledge that this is an ambitious goal for PES implementation, but we attempted to identify the overlap between distribution of areas that have high potential for provision of ecosystem services and areas of importance for carnivore conservation.

## 2.4. Protected areas versus payments for ecosystem services role conserving connectivity areas

The analyses above enabled us to identify the potential role of PES if applied extensively in the Caribbean region. We sought to compare the effect of further conservation efforts in maintaining species' connectivity areas within area-constrained scenarios. We tested whether PES or PAs conserve a greater amount of connectivity areas if each were implemented to cover the same total area.

For comparison with PES, we used two recently proposed PAs that have been selected following a systematic conservation planning approach in the region based on ecosystem and biome representativity and gaps that together sum 8610 km$^2$ (Serranía de San Lucas-SDSL and Serranía de Perijá-SDP) [36] To ensure the comparison was on an even footing, we then implemented PES by randomly allocating 1 km$^2$ pixels that have been identified as 'hotspots' within the prioritization developed by Rodríguez and collaborators [88] to the equivalent area of the proposed PAs ($N$ = 8610 km$^2$). We did not use prioritization algorithms for PA selection because the Colombian legislation does not currently require it.

## 2.5. Data sources

We collected information on species occurrences, cover types used as habitat, home range sizes and dispersal distances for all mammalian species of the order Carnivora that have been reported in the Caribbean region of Colombia. For the total of 22 species, we gathered all available occurrences derived from four primary sources: (i) biodiversity datasets (Global Biodiversity Information Facility, GBIF.org), (ii) published peer-reviewed papers, (iii) available dissertations and technical reports, and (iv) existing databases from the region (Sistema de Información sobre Biodiversidad de Colombia, and the Instituto de Ciencias Naturales, Universidad Nacional de Colombia). Species taxonomic identification and accuracy of occurrence records were validated using the most current taxonomic list of mammals in the country and the region, and were checked by regional mammal experts [49,89]. Since four species had fewer than 20 independent occurrence records, we eliminated these from the analysis (i.e. *Speothos venaticus*, *Bassaricyon alleni*, *Bassaricyon neblina* and *Bassaricyon medius*). We removed two other species due to low accuracy of the model results (i.e. *Nasua nasua*, *Lontra longicaudis*). We present detailed information for each species in electronic supplementary material, S1.

Information on home range sizes and habitat associations was gleaned from an extensive Web of Science search, previous reviews of all mammalian carnivores [74,75] and IUCN Red List assessments. For the Web of Science search, we used as criteria the scientific and common name of each species (both in English and Spanish), in combination with each of the following words: 'habitat', 'connectivity', 'home range', 'dispersal', 'matrix' and 'resistance'. We recorded all land-cover types reported as habitat for each species and compared and validated them using previous ecological reviews of the group and the existing national knowledge (electronic supplementary material, S1 and S3) [49,74,75,90]. We also recorded the home range size reported for each species, the associated country of the study, and the reported method used for calculation. Reports of species home range sizes varied across countries and methods of calculation. We therefore used the median home range value reported for each species after removing extreme outliers; we used the remove outliers function based on Tukey's method, to identify the outliers ranging above and below the 1.5 interquartile range approach (IQR). To calculate species' dispersal distances, we used the median home-range size reported for each species and applied the well-known isometric relationship between dispersal distance and home-range size [91] (electronic supplementary material, S3).

For ecological niche models, we used only scenopoetic, non-dynamically linked environmental variables [70]. We used 19 climatic variables obtained from WorldClim v. 2 [92]. To identify suitable patches, we used the land-cover types from the official Colombian national cartography, which follows the Corin Land Cover classification scheme [93,94]. This layer was made using 2007 Landsat imagery at a scale of 1 : 100 000. The maps of suitable habitat patches were created using ArcGIS 10. We used the Digital Elevation Model derived from the ASTER GDEM [95] to identify the elevation range for each species. To define biogeographic provinces, we used the national cartography constructed by the Colombian Environmental Ministry and the National Parks Unit [96].

Our approach uses the best available existing information but we acknowledge that due to the history of conflict in Colombia, and inadequate financial resources devoted to inventory and monitoring, biodiversity information is necessarily scarce. In this context, it was necessary to make subjective decisions (e.g. in relating published habitat associations to national land-cover categories). However, to maximize reproducibility of this analysis, we have reported each of the variables used for each species in the electronic supplementary material, including the information used to select habitat types, home range sizes, and dispersal distance analysis, detailed information of covariates used for niche modelling, spatial representations of niche and habitat patches, spatial representation of species-specific resistances, and spatial representation of priority patches (electronic supplementary material, S1, S2, S3).

# 3. Results

## 3.1. Habitat identification

Based on the presented approach for habitat identification, we found that the potential distributions of the 16 species of carnivores considered covered the vast majority of the Caribbean region (mean = 71.4% ± 28 s.d., range: 12.9–93.2%). However, these climatically suitable areas were substantially reduced when we filtered potential distributions to include only areas within suitable habitat patches (mean = 27.4% ± 18.9 s.d., range: 6.8–87.1%). The difference in these percentages reflects both habitat loss for the most generalist species (e.g. *P. concolor* and *M. frenata*) and restricted habitat use for the more specialized species (e.g. *L. tigrinus*, *N. olivacea* and *T. ornatus*) (figure 3).

Of the biogeographic provinces examined, we found that the Northern Andean province contains the highest percentage of mammalian carnivores' habitat (mean = 35.2 ± 13.9 s.d., range: 22.4–69.4%) with the Choco-Magdalena-Catatumbo province as a close second (mean = 34.7 ± 12.2, range: 0–54.8%). Two species have their potential distributions and habitat entirely constrained to the two mountainous biogeographic provinces of the region (*L. tigrinus* and *N. olivacea*), which largely reflects the altitudinal limits of their distributions.

Using the PAs limits, we found that most of the suitable habitat patches for all species occur outside of existing PAs (mean = 78.1 ± 11.1 s.d., range: 51–90.8%). As expected, most remaining suitable habitat patches occur in mountain landscapes (mean = 53.3 ± 19.0 s.d., range: 31.5–95.6%), which reflects lower development and land-use change due to geographic isolation, as well as reduced access due to the presence of long-term conflict.

Using the Global Forest Watch information, we also identified forest cover loss within the identified habitat from 2007 to 2016. Over the 9-year time period, mean loss of connectivity areas was 1875 km$^2$ (1430 ± 79 s.d., range: 81–5910 km$^2$) which is an average of 2.6% of remaining connectivity area per species over the full-time period (±0.8 s.d., range: 0.7–3.5%) (figure 4). The mean annual forest loss within the suitable habitat for all species combined was 208 km$^2$ (±159 s.d., range: 9–657 km$^2$) which represents a mean annual loss of 0.3% of the total suitable habitat (±0.1 s.d., range: 0.1 and 0.4%); however, in 2016, this rate increased nearly twofold, which may reflect a decreasing reluctance for land developers (i.e. agriculture, mining) to conduct activities in formerly rebel-occupied areas.

## 3.2. Identification of connectivity areas

The total suitable habitat identified as connectivity area ranged between 51.7 and 97.8% across species (mean = 59.1 ± 10.9 s.d.). Connectivity areas covered 17.8% (18.8 s.d.) of the study area and ranged from 5694 km$^2$ for *L. tigrinus* (2.9% of the total study area) to 163 873 km$^2$ for *C. thous* (84.1% of the total study area). Most connectivity areas occur outside of existing PAs (mean 72.5 ± 17.5 s.d., range: 30.5–86.2%) and are located in mountain landscapes (mean = 54.1 ± 30.4–99.4 s.d., range: 31.5–95.6%). For species included on the IUCN Red List (with threat categories of threatened or greater) or the national endangered species listing in any threat category, only 27 000 km$^2$ area remained in identified connectivity areas (less than 13% of the entire study area).

## 3.3. Protected areas representativity and the potential role of payments for ecosystem services

Neither current PAs or potential PES areas successfully cover all species' connectivity areas. Currently, only a small proportion of connectivity areas are actively protected by the existing PAs (mean = 26.8 ± 18 s.d., range: 9–69.4%). There is great potential for PES to conserve areas of habitat connectivity importance if applied extensively (mean = 45.4 ± 12.8 s.d., range: 23.9–64.8%) (figure 5). PAs

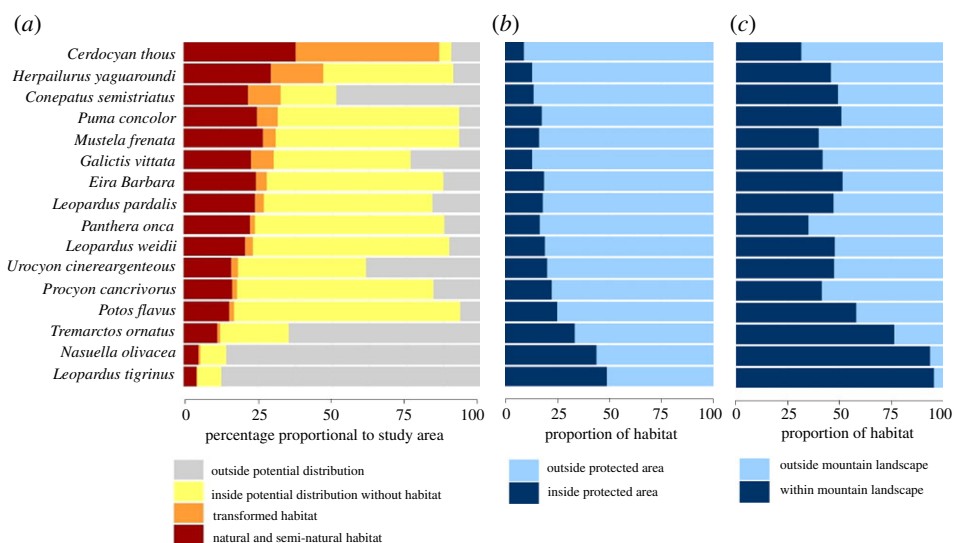

**Figure 3.** (*a*) Percentage of species potential distribution area within the study region and proportion of transformed or natural and semi-natural cover types area within the habitat of each species. In (*a*), 100% corresponds to the total area of the Caribbean region. (*b*) Proportion of habitat area within PAs for each species. (*c*) Proportion of habitat area within mountain landscapes for each species. In (*b*,*c*), 100% corresponds to the total area of suitable habitat in the Caribbean region for each species. Note that the majority of habitat for all species falls outside of existing protected areas and a substantial proportion occurs in remaining intact forest in mountain landscapes.

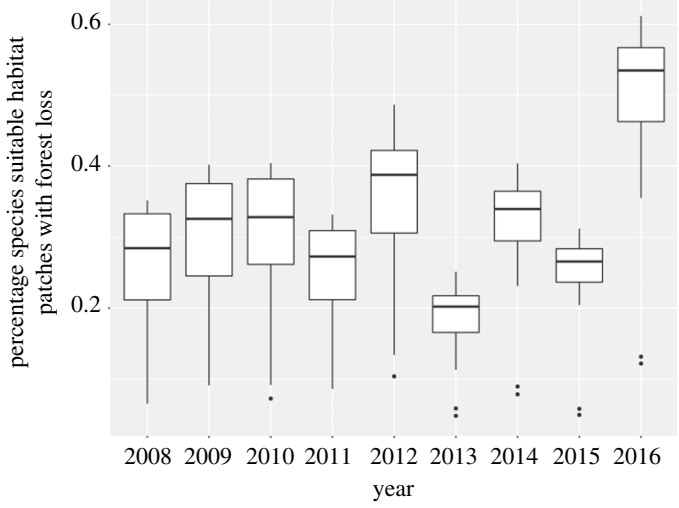

**Figure 4.** Percentage of area within the identified habitat that faced forest loss for all the species between 2008 and 2016. In this analysis, 100% corresponds to the total area of habitat of each species.

nevertheless play an essential role in safeguarding habitat connectivity areas, and are particularly important for species that are constrained to mountain landscapes. Examples include *L. tigrinus* with 69.43% and *N. olivacea* with 69.24% of their connectivity areas already within the existing PAs (figure 5).

Even if Colombian environmental agencies extensively implement PES schemes, a significant portion of the important connectivity areas would require alternative habitat conservation strategies, given that they fall outside of either PAs or PES (mean = 27.8 ± 15.9 s.d., range: 5.9–62.6%). This is particularly pronounced for species that use a broader set of cover types as habitat (*C. thous*, *U. cinereargentous*, *P. onca*) (figure 5).

## 3.4. PA versus PES roles in conserving connectivity areas

The mean percentage of connectivity areas currently covered by PAs is around 27.1%. Increasing PAs using potential new PAs (sum together 8610 km$^2$ representing an increase of less than 5% of the

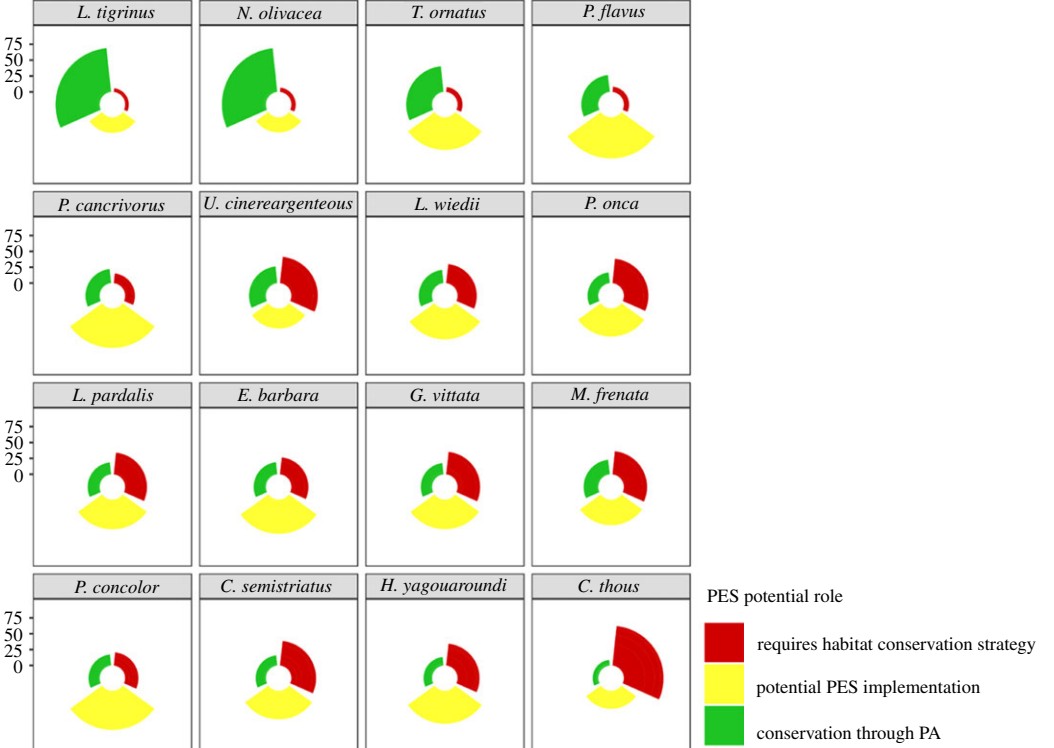

**Figure 5.** Percentage of management guidelines that can be used to ensure conservation of the species connectivity areas within the Caribbean region of Colombia. The y-axis shows percentage of area and the colours represent the potential role that a PES scheme could have conserving connectivity area for each species. This figure represents the diverse impact that different landscape conservation strategies have in different species.

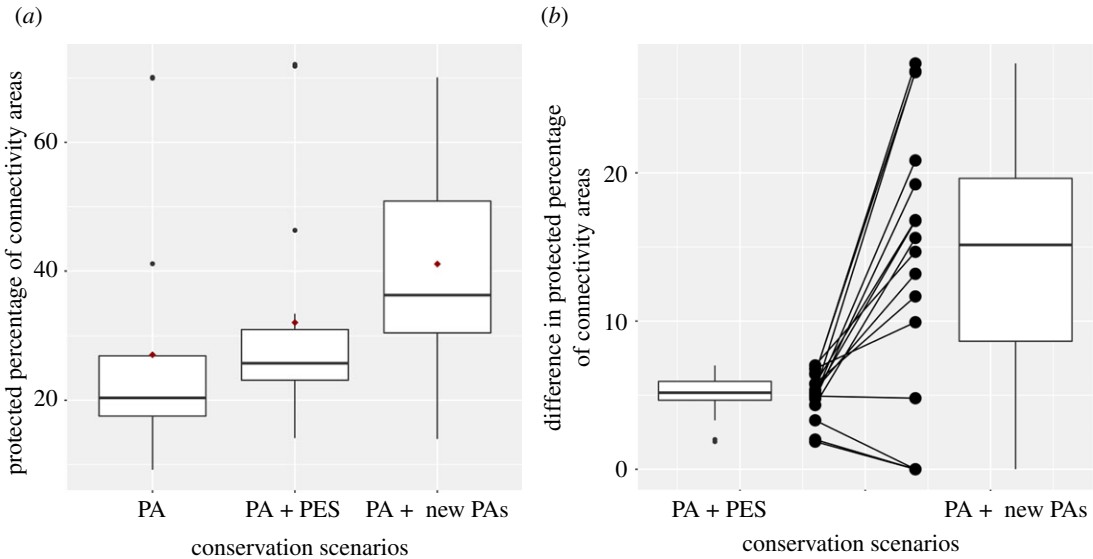

**Figure 6.** (a) Protected percentage of connectivity areas for all the species and (b) the difference on the protected percentage of connectivity areas for all the species after including the two conservation scenarios (payment of environmental services, PES, or the increment of two new protected areas, new PES). Note that under a scenario where the same land-area is attributed to both PA and PES schemes, new PAs tend to be more effective at conserving connectivity for mammalian carnivores.

existing national PAs area), increases the protection of connectivity areas from 27.1% to 41.1% (±16.9 s.d., range: 51.4–70.1%), a net gain of 14.0% (±9.3 s.d., range: 0–27.4%). By contrast, conserving the same area under a PES scenario increases conservation of connectivity areas to 32.1% (±17.0 s.d., range: 14.1–72.1%) an incremental gain of only 5.0% (±1.5 s.d., range between 1.9 and 7.1 per cent) (figure 6).

When comparing the two conservation approaches, which together represent the full suite of current policy instruments in Colombia, we found that 12/16 species have a higher percentage of connectivity areas covered by implementing new PAs than using a PES approach (figure 6). Arboreal species such as *L. weidii* or *P. flavus* benefit disproportionately from the PA approach, gaining as much as 27% conserved connectivity area over a PES strategy. Endangered species such as *P. onca* and *T. ornatus* also stand to gain substantially under the PA scenario (19.2 and 26.8%, respectively). On the other hand, three species do not gain any connectivity area if new PAs are created (*L. tigrinus*, *N. olivacea* and *U. cinereargenteus*); two of those listed in the national and international threatened species lists.

# 4. Discussion

Overall, our results indicate that under a comparative scenario where area conserved in PES is equivalent to new PAs, PES was substantially less efficient at conserving connectivity areas for most of the species considered. Implementation of PAs focuses on continuous large areas and this approach is often highly effective at reducing deforestation [1,16,97]. We focused our analysis on priority connectivity patches, ensuring that patches reflect population requirements, meaning they are sufficiently large to incorporate individual territories [30]. In this respect, PAs are more likely to ensure movement between patches as they are larger than the individual home ranges of all species in our study ($x = 33.8$ km$^2$ $\pm$ 64.1 km$^2$ s.d.). By contrast, lands selected for PES tended to be smaller than the home ranges of many species we considered (mean forest size = 0.15 km$^2$) [40]. Although PES is increasingly gaining international support, the efficacy of this strategy in biodiversity conservation is still not well known in many countries (e.g. Colombia) [3,98].

Overall, it is likely that the designation of new PAs could have a greater impact on halting further habitat loss and therefore warrant short-term conservation for most species. Large, contiguous reserves are also more resilient to direct anthropogenic pressures such as bushmeat hunting [99]. However, such reserves will require greater long-term economic support from government funds, which is unlikely in Colombia where funding is scarce [100]. PES implementation often results in conservation of smaller forest fragments that are likely to be more vulnerable to these harder-to-monitor drivers of biodiversity loss. On the other hand, an efficient PES scheme should both conserve and generate revenue for local communities, which can have greater social impact and lower direct financial cost to the government over the long term. This assumes that PES is designed with clear transaction rules that allow continued contribution to local livelihoods [39]. Unfortunately, to date, quantitative comparisons of economic and social costs versus conservation benefits of PES in comparison with PAs has not yet occurred in Colombia. This is perhaps unsurprising given that such calculations must take into account the broad array of PES schemes, most of which have not been evaluated for social and economic impacts [98].

Nevertheless, we found that if applied to its full potential (as proposed by Rodríguez *et al.* [41,88]), PES implementation in the Caribbean region could have substantial conservation benefits for mammalian carnivores, despite substantial political and economic challenges [3]. We acknowledge that an extensive implementation of PES may be unlikely in the short term, but there is an ongoing discussion about the global distribution of PES schemes and payments. Therefore, the attribution of 27% of the area of Colombia under PES schemes seems feasible [100,101]. One of our major findings is that areas with high potential for provision of ecosystem services could align with areas of importance for carnivore conservation.

We also identified spatial congruence between the existing landscape conservation policy instruments and habitat and connectivity areas [10,88]. While the proposed new PAs will certainly increase conservation of connectivity areas for most of the species, the implementation of PES will require priority-setting efforts to achieve similar outcomes. Such priority setting should incorporate habitat distributions and connectivity information. Our analysis did not provide a multi-species weighted prioritization, as our objective was not to provide a single solution for multiple species. The main objective of this analysis was to provide evidence of the potential conservation outcomes of different landscape conservation decisions on several species with different spatial requirements. This approach enables quantification of the differential impacts of the two approaches over a diverse but phylogenetically related species group. Nevertheless, we did find that priority patches overlapped for many species. These spatial results could be included as a conservation target in a spatial prioritization analysis to further decide where to create new PAs or other effective area-based conservation measures [33,57,101]. Our findings could also be used in surrogate species approaches

when stakeholders require species-specific management strategies, as is the case with several endangered species [102,103].

Interestingly, our results support the use of both policy instruments [10,88] but reaffirm the urgent need in Colombia to protect areas previously identified by the national systematic conservation planning efforts [10]. It will be critical to support the creation of the two currently proposed PAs (Serranía de San Lucas and Serranía de Perija) since these two areas house most of the remaining unprotected natural ecosystems of the region [10]. Our results indicate that these areas will have substantial benefits for habitat conservation and connectivity for most carnivore species [104–106]. Nevertheless, to conserve all carnivore connectivity areas and target species-specific needs, it will be crucial to develop new policies that target species' habitat and populations. This should include species-specific requirements which are quite different depending on the distribution and ecological requirements of each species. This is evident when comparing high altitude species (*L. tigrinus*, *N. olivacea*), or species with high forest vertical structural complexity requirements (*P. flavus*) agains generalist species (*C. thous*, *P. concolor*). In each of these species, the impact of each conservation decision varies as its potential impacts conserving connectivity areas. Effective conservation will probably require management in areas that fall outside of existing PAs or other existing policy instruments [55,85,107]. Particular emphasis should be given to endangered species; new legislation is required that acknowledges the importance of understanding species-specific habitat and connectivity requirements [55,108,109].

The impact of habitat loss and fragmentation on carnivores has been shown worldwide, particularly in biodiversity-rich tropical countries [21,26]. Conserving continuous habitat will be particularly important for wide-ranging mammal species such as carnivores [25,110], and for forest specialist species that have less capacity for dispersal through the matrix between forest patches [110]. However, ecological information on species' habitat requirements in tropical countries is generally poor [74,90]. Sparse existing information is based on global analyses that typically rely upon coarse-resolution data [42,44,111,112] that is inadequate for regional-scale planning. Unfortunately, regional and local-scale analyses are scarce for mammalian species from the order Carnivora; this is true even for the most charismatic and well-studied species [90,102].

We found that the amount of remaining suitable habitat for carnivore species of the region is very small in comparison with the total potential distribution. Except for some of the more generalist species (*C. semistriatus*, *H. jaguarondi* and *C. thous*), most of the carnivore species have less than half of their potential distribution remaining in suitable habitat. We also found that habitat and connectivity areas are distributed primarily in mountain landscapes, most likely due to restricted human access and therefore lower potential for development [50,113,114]. Additionally, our results prioritize large patches; such patches are well known to be critical for maintaining metapopulation dynamics and being more resistant to anthropogenic impacts [21,115–117]. Further analysis should examine the influence of other smaller patches and their potential as stepping stones for connectivity [118,119].

To our knowledge, this is the first evaluation of conservation policy instruments in Colombia that uses habitat connectivity as a criterion. This study is also the first species-specific connectivity conservation modelling effort undertaken for an entire mammalian order. Previous studies have either been conducted at a coarser global scale [41,44,85,120], focused only on a few species of carnivores [121–123], or have targeted only habitat distributions, without including connectivity or spatial requirements [121,124].

We expect that our overall approach to using carnivores as a focal taxa will be useful in future studies, especially in data-poor regions where information on less common species is lacking, and particularly for community-based conservation schemes [33,82]. Previous efforts in Colombia, at least for PES, are starting to use carnivores as flagships for such efforts [37]. Furthermore, a number of studies have proposed that carnivores may represent the needs of other biodiversity groups [125]. Also, our methodological approach provides a framework for identification and prioritization of areas of connectivity importance, a requirement within the current guidelines of PAs management and efficiency based on the Aichi goals, including a species-specific approach that global exercises cannot fulfil [126].

# 5. Conclusion

Colombia is one of the megadiverse countries of the world, but political and social turmoil threatens to substantially increase pressure on this biodiversity [35,127]. The current shifts in land-use change following the signing of the peace treaty will only accelerate rates of habitat loss for most species [128,129]. Further contributors to future biodiversity losses include biodiversity data scarcity and limited funding for science and conservation efforts [41,88,130].

PES and PAs constitute the two main conservation tools currently available in the country. Here, we report that for mammalian carnivores at least, PAs have a greater positive impact, but appropriately planned PES schemes could significantly contribute to conservation needs. The increase in PAs can substantially contribute to the protection of habitat connectivity areas while increasing ecosystem representivity and providing environmental services. Future implementation of PES can have an important potential impact, but requires regional planning to ensure a selection of private lands that enable co-benefits for environmental services and biodiversity targets. In such future analyses, it will be crucial to include habitat and connectivity needs of mammalian carnivores, and ideally other taxa. The level of spatial congruence observed in our analyses between existing landscape conservation policy instruments and connectivity areas represents additional support for previously identified new PAs (Serranía de San Lucas and Serranía de Perijá) [10,106].

We developed a spatial analysis that allows the identification of conservation needs in a species-specific framework for a data-poor country. We also created an output that can be aligned with national and regional geographical information, as it follows the national land-use and land-cover typology [55,94]. Although we acknowledge our approach requires validation, and extensive research to reduce uncertainty, the effort is useful in that it uses detailed, regional biodiversity data for landscape planning. Species-specific analysis has substantial benefits in terms of quantifying landscape-scale anthropogenic effects on biodiversity [28,131], something which has historically been overlooked in developing countries like Colombia. Previous global analysis has used less detailed habitat definitions, which we think has been detrimental to habitat conservation at the regional scale [55,102]. The use of simplified existing information from global analysis can both over- or sub-represent the actual distribution of a species, and can overlook endangered subpopulations [55,102].

In future landscape planning efforts, we recommend that other aspects of carnivore conservation, particularly considerations relating to human–wildlife conflict, be included. Landscape conservation facilitates carnivore movement [85,132,133] which can generate conflict, especially in areas of high human density, heavy agricultural development or areas that are hunted intensively [44,134].

Finally, we recommend that future prioritization schemes, both for PA establishment or PES schemes, incorporate criteria that reflect not only the habitat distributions of species but also the connectivity between these habitat patches. The current more ad hoc approach to funding and designating PES could be greatly improved by considering multiple biodiversity criteria. This could lead to prioritizing PES in either larger farms, groups of continuous private lands, communal territories, or strategically located patches that substantially increase connectivity. Such approaches could be coupled with strategies that seek poverty alleviation such as REDD+, allowing greater degrees of community and institutional empowerment [12,135,136].

Data accessibility. Original datasets and geographical datasets are included as electronic supplementary material [137].
Authors' contributions. D.A.Z.C.: conceptualization, data curation, formal analysis, investigation, methodology, validation, visualization, writing—original draft, writing—review & editing; J.F.G.-M.: conceptualization, methodology, validation, visualization, writing—original draft, writing—review & editing; A.A.-A.: conceptualization, data curation, investigation, methodology, writing—review & editing; J.S.J.-A.: data curation, investigation, resources; J.D.R.A.: data curation, investigation, resources; D.A.: conceptualization, investigation, methodology, resources, supervision, writing—original draft, Writing—review & editing; M.G.B.: conceptualization, investigation, methodology, supervision, validation, writing—original draft, writing—review & editing.

All authors gave final approval for publication and agreed to be held accountable for the work performed therein.
Competing interests. We declare we have no competing interests.
Funding. Grants from Colciencias and Fulbright supported this research.
Acknowledgments. We thank Catalina Moreno and I. Mauricio Vela-Vargas and all the staff at ProCAT Colombia for their help in the initial stages of this project. We also thank the Betts Landscape Ecology Laboratory for their contribution and comments and especially to Urs Kormann for its invaluable help during the manuscript graphics creation, and Ben Phalan and Clinton Epps for their insights that helped to improve the paper.

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
