## [Peer Review File · Royal Society Open Science]

Review History

RSOS-201154.R0 (Original submission)

Review form: Reviewer 1

Is the manuscript scientifically sound in its present form?

Yes

Are the interpretations and conclusions justified by the results?

Yes

Is the language acceptable?

Yes

Do you have any ethical concerns with this paper?

No

Have you any concerns about statistical analyses in this paper?

No

Recommendation?

Accept with minor revision (please list in comments)

Comments to the Author(s)

Connectivity conservation at the crossroads: protected areas vs payments for ecosystem services in conserving connectivity for Colombian carnivores; Zárrate-Charry et al.; Royal Society Open Science

I read with great interest the manuscript titled “Connectivity conservation at the crossroads: protected areas vs payments for ecosystem services in conserving connectivity for Colombian carnivores” by Zárrate-Charry et al. In this study, the authors focus on a set of terrestrial carnivore species in Colombia’s Caribbean region, assess their potential connectivity patterns, identify areas important for their movement, and finally examine the spatial congruence between locations with high potential connectivity for all species and PAs versus PES areas.

I submit that it was a pleasure reading this manuscript. The authors have done a commendable job, not only in terms of the research, but also in the way the results have been disseminated. The manuscript is very well written (particularly the lucid description of methods), and the pitch/narrative makes a convincing case for how relevant this study could be for conservation planning in Colombia, moving forward.

I have very few minor comments and some queries. If the authors could make these changes and provide clarifications, I would be happy to see this study published in RSOS.

Introduction:

(Note- page numbers here refer to the original numbers in the submitted MS)

Page 2 L22: I suggest removing “such as Colombia” here. The paragraph is generally applicable to most countries in the tropics, and certainly holds true almost all countries in the neo-tropics.

Page 3: Consider adding a line or two in the last paragraph about why you chose carnivores as the focal species. Wide-ranging, long dispersal distances, high degree of endangerment etc.

Materials and Methods:

Page 3 L47–50: Recently there was a data paper that presented a combined database of all neotropical mammals. I wonder if the authors used this information as well and/or were part of that exercise.

Page 4 L22–27: I really appreciate that the authors are being forthright about this. This would entail some subjectivity, but is justified given the general scarcity of information from a lot of biodiversity-rich countries that don’t have a long history of systematic monitoring of wildlife and habitats.

Page 5 L54–57: Please consider presenting a table in the main MS, with a list of species against all habitats considered, and the corresponding resistance value used for each species.

Discussion:

I have only two suggestions for this section.

1. The analysis did not involve “weighting” of individual species, i.e., endangered or rare species, or those that were more vulnerable to demographic effects of metapopulation level disruptions were not given higher weights while using the combined connectivity areas. The authors should deliberate briefly on if and how this would alter their results; or, if similar studies are carried out using a wider set of species, how would incorporating these weights improve the assessments. There’s quite a bit of spatial prioritization literature that the authors can look into for this.

2. A key component that I felt was missing in the Discussion was some comparison of economic or social costs of declaring PAs versus implementing PES. I fully understand that it was not the main focus of the analysis. Since the authors propose that both approaches may be useful, but PAs would definitely benefit some species better, it would be worthwhile to present some information on the financial costs and/or social costs of both these initiatives.

Figures: The captions need not start with “Boxplot showing...”. You can directly describe the figure without saying it’s a boxplot in every instance.

Finally, I suggest the authors thoroughly proofread the manuscript for typographical and grammatical errors.

Review form: Reviewer 2 (Philip Riordan)

Is the manuscript scientifically sound in its present form?

No

Are the interpretations and conclusions justified by the results?

No

Is the language acceptable?

Yes

Do you have any ethical concerns with this paper?

No

Have you any concerns about statistical analyses in this paper?

Yes

Recommendation?

Major revision is needed (please make suggestions in comments)

Comments to the Author(s)

See attached document (Appendix A).

Decision letter (RSOS-201154.R0)

Dear Dr Zárrate-Charry

The Editors assigned to your paper RSOS-201154 "Connectivity conservation at the crossroads: protected areas vs payments for ecosystem services in conserving connectivity for Colombian carnivores" have now received comments from reviewers and would like you to revise the paper in accordance with the reviewer comments and any comments from the Editors. Please note this decision does not guarantee eventual acceptance.

Please submit your revised manuscript and required files (see below) no later than 21 days from today's (ie 03-Feb-2021) date. Note: the ScholarOne system will 'lock' if submission of the revision is attempted 21 or more days after the deadline. If you do not think you will be able to meet this deadline please contact the editorial office immediately.

on behalf of Dr Agnieszka Latawiec (Associate Editor) and Pete Smith (Subject Editor)
openscience@royalsociety.org

Associate Editor Comments to Author (Dr Agnieszka Latawiec):

Associate Editor: 1

Comments to the Author:

Dear Authors,

Please carefully incorporate the comments of both reviewers, with a special attention to the second reviewer (separate document is attached with details questions and suggestions).

Kind Regards,
Agnieszka Latawiec

Reviewer comments to Author:

Reviewer: 1

Comments to the Author(s)

Connectivity conservation at the crossroads: protected areas vs payments for ecosystem services in conserving connectivity for Colombian carnivores; Zárrate-Charry et al.; Royal Society Open Science

I read with great interest the manuscript titled “Connectivity conservation at the crossroads: protected areas vs payments for ecosystem services in conserving connectivity for Colombian carnivores” by Zárrate-Charry et al. In this study, the authors focus on a set of terrestrial carnivore species in Colombia’s Caribbean region, assess their potential connectivity patterns, identify areas important for their movement, and finally examine the spatial congruence between locations with high potential connectivity for all species and PAs versus PES areas.

I submit that it was a pleasure reading this manuscript. The authors have done a commendable job, not only in terms of the research, but also in the way the results have been disseminated. The manuscript is very well written (particularly the lucid description of methods), and the pitch/narrative makes a convincing case for how relevant this study could be for conservation planning in Colombia, moving forward.

I have very few minor comments and some queries. If the authors could make these changes and provide clarifications, I would be happy to see this study published in RSOS.

Introduction:

(Note- page numbers here refer to the original numbers in the submitted MS)

Page 2 L22: I suggest removing “such as Colombia” here. The paragraph is generally applicable to most countries in the tropics, and certainly holds true almost all countries in the neo-tropics.

Page 3: Consider adding a line or two in the last paragraph about why you chose carnivores as the focal species. Wide-ranging, long dispersal distances, high degree of endangerment etc.

Materials and Methods:

Page 3 L47–50: Recently there was a data paper that presented a combined database of all neotropical mammals. I wonder if the authors used this information as well and/or were part of that exercise.

Page 4 L22–27: I really appreciate that the authors are being forthright about this. This would entail some subjectivity, but is justified given the general scarcity of information from a lot of biodiversity-rich countries that don’t have a long history of systematic monitoring of wildlife and habitats.

Page 5 L54–57: Please consider presenting a table in the main MS, with a list of species against all habitats considered, and the corresponding resistance value used for each species.

Discussion:

I have only two suggestions for this section.

1. The analysis did not involve “weighting” of individual species, i.e., endangered or rare species, or those that were more vulnerable to demographic effects of metapopulation level disruptions were not given higher weights while using the combined connectivity areas. The authors should deliberate briefly on if and how this would alter their results; or, if similar studies are carried out

using a wider set of species, how would incorporating these weights improve the assessments. There's quite a bit of spatial prioritization literature that the authors can look into for this.

2. A key component that I felt was missing in the Discussion was some comparison of economic or social costs of declaring PAs versus implementing PES. I fully understand that it was not the main focus of the analysis. Since the authors propose that both approaches may be useful, but PAs would definitely benefit some species better, it would be worthwhile to present some information on the financial costs and/or social costs of both these initiatives.

Figures: The captions need not start with "Boxplot showing...". You can directly describe the figure without saying it's a boxplot in every instance.

Finally, I suggest the authors thoroughly proofread the manuscript for typographical and grammatical errors.

Reviewer: 2

Comments to the Author(s)

See attached document.

===PREPARING YOUR MANUSCRIPT===

===PREPARING YOUR REVISION IN SCHOLARONE===

Author's Response to Decision Letter for (RSOS-201154.R0)

See Appendix B.

RSOS-201154.R1 (Revision)

Review form: Reviewer 1

Is the manuscript scientifically sound in its present form?

Yes

Are the interpretations and conclusions justified by the results?

Yes

Is the language acceptable?

No

Do you have any ethical concerns with this paper?

No

Have you any concerns about statistical analyses in this paper?

No

Recommendation?

Accept with minor revision (please list in comments)

Comments to the Author(s)

Connectivity conservation at the crossroads: protected areas vs payments for ecosystem services in conserving connectivity for Colombian carnivores; Zárrate-Charry et al.; Royal Society Open Science; Review #2

I believe the authors have adequately addressed all the major comments raised by me and the other reviewer in the previous round. I have a few minor comments that authors need to make before this paper can be accepted for publication.

Abstract:

L27-28: This line says 16 of 22 carnivores but elsewhere in text the authors say 16 of 24. Please correct as necessary.

Introduction:

In trying to incorporate reviewer comments, I think the flow/narrative of the text has become a bit distorted. The authors have done a good job in including information that was missing from the previous version. But I suggest reworking the text in this section to make it more streamlined. For example, Page 2 L17-24 should come after Page 2 L51. Similarly, I think the authors should try and structure the paragraphs such that they are thematically linear (each para should address 1-2 key themes; start with broader picture and narrow down to Colombia and carnivores).

Methods:

Again, please split paragraphs carefully such that each para has 1-2 central themes with relevant citations.

Page6 L18: "Aichi"

Results:

All subheadings are labeled "4.1". Please correct these.

Page8 L10, L20, L46: It would be better to start these paragraphs by adding one line that reiterates what aspect was being examined and how. Example, "Based on potential distribution of species mapped using climatic variables under the MaxEnt framework, we found...";

Page9 L24: I suggest providing a more informative sub-heading.

Discussion:

Same comment here as with the Introduction. Please restructure/reorganize the text so that the narrative is coherent and linear. Start with the key insights from your analysis, move on to highlight species-specific insights, then discuss the broader issues with PAs versus PES schemes in Colombia, draw parallels to research from other landscapes or locations, add 1-2 paragraphs of limitations, and finally a short, may be 1 or 2 paragraphs of conclusions.

Figure 5: These results are interesting and should be discussed in some more detail (see my comment above on species-specific insights in the Discussion).

Finally, I still found a lot of typographical and grammatical errors throughout the manuscript (missing periods, commas or other punctuation marks, spelling errors, sentences starting with a small letter, etc.). Some sentences are also framed awkwardly and need to be rewritten. I understand that the MS was already proofread by someone who was not involved in the study, but I encourage the authors to make edits as suggested above and seek assistance/guidance with language once again.

Decision letter (RSOS-201154.R1)

Dear Dr Zárrate-Charry

On behalf of the Editors, we are pleased to inform you that your Manuscript RSOS-201154.R1 "Connectivity conservation at the crossroads: protected areas vs payments for ecosystem services in conserving connectivity for Colombian carnivores" has been accepted for publication in Royal Society Open Science subject to minor revision in accordance with the referees' reports. Please find the referees' comments along with any feedback from the Editors below my signature.

Please submit your revised manuscript and required files (see below) no later than 7 days from today's (ie 17-Sep-2021) date. Note: the ScholarOne system will 'lock' if submission of the revision is attempted 7 or more days after the deadline. If you do not think you will be able to meet this deadline please contact the editorial office immediately.

on behalf of Dr Agnieszka Latawiec (Associate Editor) and Pete Smith (Subject Editor)
openscience@royalsociety.org

Reviewer comments to Author:

Reviewer: 1

Comments to the Author(s)

Connectivity conservation at the crossroads: protected areas vs payments for ecosystem services in conserving connectivity for Colombian carnivores; Zárrate-Charry et al.; Royal Society Open Science; Review #2

I believe the authors have adequately addressed all the major comments raised by me and the other reviewer in the previous round. I have a few minor comments that authors need to make before this paper can be accepted for publication.

Abstract:

L27-28: This line says 16 of 22 carnivores but elsewhere in text the authors say 16 of 24. Please correct as necessary.

Introduction:

In trying to incorporate reviewer comments, I think the flow/narrative of the text has become a bit distorted. The authors have done a good job in including information that was missing from the previous version. But I suggest reworking the text in this section to make it more streamlined. For example, Page 2 L17-24 should come after Page 2 L51. Similarly, I think the authors should try and structure the paragraphs such that they are thematically linear (each para should address 1-2 key themes; start with broader picture and narrow down to Colombia and carnivores).

Methods:

Again, please split paragraphs carefully such that each para has 1-2 central themes with relevant citations.

Page6 L18: "Aichi"

Results:

All subheadings are labeled "4.1". Please correct these.

Page8 L10, L20, L46: It would be better to start these paragraphs by adding one line that reiterates what aspect was being examined and how. Example, "Based on potential distribution of species mapped using climatic variables under the MaxEnt framework, we found...";

Page9 L24: I suggest providing a more informative sub-heading.

Discussion:

Same comment here as with the Introduction. Please restructure/reorganize the text so that the narrative is coherent and linear. Start with the key insights from your analysis, move on to highlight species-specific insights, then discuss the broader issues with PAs versus PES schemes in Colombia, draw parallels to research from other landscapes or locations, add 1-2 paragraphs of limitations, and finally a short, may be 1 or 2 paragraphs of conclusions.

Figure 5: These results are interesting and should be discussed in some more detail (see my comment above on species-specific insights in the Discussion).

Finally, I still found a lot of typographical and grammatical errors throughout the manuscript (missing periods, commas or other punctuation marks, spelling errors, sentences starting with a small letter, etc.). Some sentences are also framed awkwardly and need to be rewritten. I understand that the MS was already proofread by someone who was not involved in the study, but I encourage the authors to make edits as suggested above and seek assistance/guidance with language once again.

===PREPARING YOUR MANUSCRIPT===

If you have been asked to revise the written English in your submission as a condition of publication, you must do so, and you are expected to provide evidence that you have received language editing support. The journal would prefer that you use a professional language editing

service and provide a certificate of editing, but a signed letter from a colleague who is a native speaker of English is acceptable. Note the journal has arranged a number of discounts for authors using professional language editing services (<https://royalsociety.org/journals/authors/benefits/language-editing/>).

===PREPARING YOUR REVISION IN SCHOLARONE===

<https://royalsociety.org/journals/authors/author-guidelines/#supplementary-material> to

include a suitable title and informative caption. An example of appropriate titling and captioning may be found at https://figshare.com/articles/Table_S2_from_Is_there_a_trade-off_between_peak_performance_and_performance_breadth_across_temperatures_for_aerobic_sc_ope_in_teleost_fishes_/3843624.

Author's Response to Decision Letter for (RSOS-201154.R1)

See Appendix C.

Decision letter (RSOS-201154.R2)

Dear Dr Zárrate-Charry,

I am pleased to inform you that your manuscript entitled "Connectivity conservation at the crossroads: protected areas vs payments for ecosystem services in conserving connectivity for Colombian carnivores" is now accepted for publication in Royal Society Open Science.

on behalf of Dr Agnieszka Latawiec (Associate Editor) and Pete Smith (Subject Editor)
openscience@royalsociety.org

Appendix A

Review: “Connectivity conservation at the crossroads: protected areas vs payments for ecosystem services in conserving connectivity for Colombian carnivores”. Royal Society Open Science.

The authors should firstly be applauded for their efforts to bring together such an ambitious study, particularly given the inevitable restrictions on data availability for many of the species they focus on. The volume of work here is impressive, and I am sure there are numerous novel insights emerging from this collation.

However, the authors ask a lot of the reader to trust the outputs. Whilst each step was thorough in many regards, I was frequently frustrated that vital details needed to follow the logic were often missing. Each of the datasets are treated to a number of manipulations before they are combined into the final analysis. Many of these treat each species or each dataset as equivalent, which is not necessarily the case. A more nuanced approach to data treatment prior to modelling, perhaps allied to simpler modelling approaches, particularly where sample sizes are small, would make this all the stronger. Where arbitrary cut-off values have been used, it would be helpful to show sensitivity analyses in subsequent models to understand the relative impacts of small changes to each derived parameter value.

I agree entirely with the author’s comments that issues of data scarcity, particularly in developing countries, should not prevent attempts to draw out important information that can assist with efforts to protect biodiversity. As I say, I can’t help feeling that this paper would have been stronger if a more simple approach had been taken. This often appears somewhat overly complex, with no strong rationale for why this complexity is required. This this sadly casts doubt on the results, which is a pity, as I am sure there is many value findings that need to be presented.

I am still not entirely clear what the comparison between the PA and PES approaches actually entails. For example, what are the differences between the two that might lead to changes in mammalian carnivore occupancy and movement? Would there be more or less suitable habitats for each species for instance and how would that come about? Given the aims of the paper, more detail on the distinction between PA and PES strategies is needed.

One of the main global threats to mammalian carnivores is conflict-driven persecution. It would be interesting to see a little more consideration of this in light of the differences between PAs and PES. PAs essentially seek to separate wildlife from people, whereas PES potentially seeks a more integrated model. It is difficult to argue livestock depredation or risk to people’s lives as being an ecosystem service worthy of payment, so the PES approach is immediately problematic for this taxon. It is arguable that any fully functioning restored or protected natural or semi-natural ecosystem will include predators as an essential part of its trophic structure, e.g. limited populations of potentially problematic prey or meso-predators. In this context, what are the goals of PES (i.e. what ecosystem services are being paid for) and what are the plausible consequences for both people and wildlife?

I offer my more detailed thoughts below. Line numbers indicated are approximate, as they don’t quite align with the text in the draft I received.

Page 3; Lines 14-20: Are PES and PA actually being presented as alternatives? As you point out, they are focused on different things, so is there evidence of PA loss as a result of enhanced PES, for example?

Page 3; Lines 53-55: What is the distribution of PES across the region? Is the 130km² contiguous or scattered? Why are these PES there? Are PES measures put in place in response to EIA requirements or some other intervention?

Page 4; Line 8: What is being proposed in the “proposed PES schemes”? It’s not clear what these are intended to do, nor whether they are reactive or proactive.

Page 4; Lines 12-13: “discussion on the expansion of both” is a little vague. It would be helpful to see more details about what is being proposed and how these two approaches are being weighed against each other.

Page 5; Lines 2-8: this is potentially a good way to incorporate a wide range of data from different sources and methodologies. However, you do not appear to have accounted for possible errors or biases between studies/methods, particularly given the differences in the number of report/papers for each species. Your approach of taking estimates and using them in subsequent calculations has the potential to further compound these errors. Would it have been possible to explore these errors a little more before settling on parameter values? For example, a limited radio tracking study of a small number of a large species should not be treated as equivalent to a larger study using GPS technology. Similarly, telemetry of all types is not necessarily equivalent for species in different habitats (open vs closed habitats for example).

Page 5; Lines 10-11: by “*scenopoetic, non-dynamically linked environmental variables*”, do you mean ‘independent’ or is there some other relationship between variables? Either way, how did you determine this? The tables in ESM1 (e.g. Table 4) do not provide sufficient information about these variables, including the selection and contribution statistics used.

Page 5; Lines 22-24: Given the inevitable discrepancies between difference land cover or habitat definitions, how did you make these “subjective decisions”? It’s good that you have provided a comprehensive comparison of habitats for each species (e.g. ESM1 Table 2), but how were these then related to the Colombian national maps from which you defined the patches?

Page 5; Lines 46-52: while these methods might overcome some of the limitations of the often small and spatially limited species occurrence data, I think it’s asking a lot in some cases. For species with the fewest points, why bother with MaxEnt at all? This is arguably making the analysis appear less plausible, given the extent to which you are pushing these limited data. Would a simple habitat definition based on published data applied to the land cover maps have sufficed? From some AUC plots (e.g. ESM1 fig. 1) there seems to be little convergence.

Page 6; Line 10: is the “minimum training presence logistic threshold” given as “MTLPT” in ESM1 (e.g. table 5)? If so, is this the threshold above which habitats were considered suitable? This needs to be clearer as for some species the MTLPT is very low (e.g. 0.07 in ESM1 Table 10). Given that this threshold appears only to use training data, doesn’t the minimum risk over-estimating the extent of suitable habitat? What was the training/test split?

Page 6; Lines 53-56: It is not clear how you derived the resistance layers for each species. From the text it appears that you used one resistance model for all species, based on the review paper by Zeller et al (2012), but I’m not sure if this is correct. Why did you not use an inverse function of the habitat suitability for resistance, on the basis that each species will move through its preferred habitats more readily than through its least preferred habitats? This would seem to define cost-pathways for each species better than an arbitrary definition based on human structures etc. These variables should have been included in species habitat suitability models, so their relative impact on each species could be explored more fully. Where these relationships are poorly understood, then a range of plausible values could be employed to look at their likely impacts, relative to other determinants of species occurrence.

Page 7: Line 24: what were the units of the three indices summed to give a connectivity rank? Were they on equal scales to prevent any index overwhelming the calculation?

Page 7: Lines 31-32: I don't follow this final sentence of the paragraph. What is 'greater habitat connectivity importance' Is it all areas/patches within the top 50% of the ranking? As you say, this is an arbitrary cut-off, so it would be helpful to see a sensitivity analysis. For example, what is the change in patch number / area if this threshold is altered?

Page 7: Lines 47-50: what is the legislation that leads you to think that these areas 'could be' PES options? This is somewhat vague, so please provide a reference.

Page 7; Lines 51-53: on what basis do you make the assumption that PES would be implemented over an area of 52,683km² (27% of the study region), particularly in light of your introduction, that states the most extensive area of PES in Colombia is 130km² ?

Pages 7 Line 56 – Page 8 Line 15: This section seems pivotal to the entire paper, in describing how PA and PES approaches were compared. In common with the introduction, I am not clear to what extent PES is actually comparable with PAs in the context presented. This clearly causes problems for the paper, so I would urge the authors to makes this clearer. What does 'we then implemented PES' in the these 1km² pixels actually mean? Was there a change in land use or land cover as a consequence, for example? What is the change that might be expected to somehow alter movement and occupancy of mammalian carnivores in these areas? How do these two proposed PAs relate to the study area in terms of comparable habitats, carnivore assemblages etc?

Page 8; Lines 42-43: If habitat suitability models did not include the human elements included in the resistance models, then perhaps it is not surprising to find habitats areas outside PAs. If this is intended to be the 'idealised' rather than the 'realised' habitat use of these species, then this needs to be clearer.

Page 8; Line 49-56: How was data from Global Forest Watch used within the connectivity analysis? I can't see any reference to it in the Methods section and figure 4 appears to show areas of loss.

Page 9; Line 32 – Are 'PEs' something different to PAs or should this be 'PES'?

Page 10; Line 7-9: This statement needs support, particularly describing more fully the PES scenario used. PES can take many different forms, so further detail is required on how representative or realistic is this PES depiction.

ESM1 Page 440 Table 30: 200 occurrences minus 29 and 2 equals 169. Why only 122 entered into model. Also, the number of parameters (52) seems quite high for the amount of data used in the model.

Appendix B

June 04, 2021
Dr. Jeremy Sanders
Editor-in-Chief
Dr. Chris Chambers
Subject Editor
Andrew Dunn
Senior Publishing Editor
Callun Shoosmith
Editorial Coordinator
Royal Society Open Science

Please find attached/enclosed the revised version of our manuscript "*Connectivity conservation at the crossroads: protected areas vs payments for ecosystem services in conserving connectivity for Colombian carnivores, Colombia*" for publication in *Royal Society Open Science*. We thank you and the reviewers for the valuable comments. We have addressed the reviewers' comments thoroughly and made changes that improved the quality of our results/our manuscript. We really appreciate your willingness to consider this revised version. Our main changes are:

1. We included a more detailed narrative to explain the PES vs PA comparison and its importance within our national context.
2. We clarified certain methodological steps and provide justifications for decisions made in each of our models.
3. We provide a more detailed description of the reasons that we used certain variables in the ecological niche and the connectivity models.

We have included a point-by-point response to the reviewers' comments below. Color codes correspond to individual changes made to the manuscript to address each reviewer's suggestions.

Thank you very much once again for this opportunity, and we look forward to your final decision.

Sincerely,

Diego Zárrate-Charry, Ph.D.
Research coordinator ProCAT-Colombia/Sierra to Sea Institute
Cellphone Colombia: (+57) 3177992028
dazarrate@procat-conservation.org, godiezcharry@gmail.com
www.procat-conservation.org

Editor: Dr Agnieszka Latawiec comments:

We have answered all the editor's comments below. The comments made by the editor or reviewers are in italics and our answers in **red font**. If our answer resulted in changes to the manuscript, we include the text in quotations.

Associate Editor Comments to Author (Dr Agnieszka Latawiec):

Associate Editor: 1

Comments to the Author:

Dear Authors,

Please carefully incorporate the comments of both reviewers, with a special attention to the second reviewer (separate document is attached with details questions and suggestions).

Kind Regards,

Agnieszka Latawiec

Response to Reviewer number 1:

Reviewer: 1

I read with great interest the manuscript titled "Connectivity conservation at the crossroads: protected areas vs payments for ecosystem services in conserving connectivity for Colombian carnivores" by Zárrate-Charry et al. In this study, the authors focus on a set of terrestrial carnivore species in Colombia's Caribbean region, assess their potential connectivity patterns, identify areas important for their movement, and finally examine the spatial congruence between locations with high potential connectivity for all species and PAs versus PES areas. I submit that it was a pleasure reading this manuscript. The authors have done a commendable job, not only in terms of the research, but also in the way the results have been disseminated. The manuscript is very well written (particularly the lucid description of methods), and the pitch/narrative makes a convincing case for how relevant this study could be for conservation planning in Colombia, moving forward.

I have very few minor comments and some queries. If the authors could make these changes and provide clarifications, I would be happy to see this study published in RSOS.

Thank you for your suggestions and comments. We made the changes proposed, made a detailed review of the document by an English native speaker, and changed some other elements to improve clarity of the document. We think your comments allowed us to improve the quality and clarity of the manuscript. We include a detailed explanation of your comments within this response letter.

Page 2 L22: I suggest removing "such as Colombia" here. The paragraph is generally applicable to most countries in the tropics, and certainly holds true almost all countries in the neo-tropics.

We removed "Such as Colombia" as proposed by the reviewer (Page 2 line 34).

Page 3: Consider adding a line or two in the last paragraph about why you chose carnivores as the focal species. Wide-ranging, long dispersal distances, high degree of endangerment etc.

We included a small paragraph addressing the importance regarding this analysis using carnivores. We included (page 3 Line 28) our comment regarding carnivore species importance "We selected this

group for three reasons. First, detrimental effect that habitat loss and fragmentation on this taxon are well known due to low densities, slow life histories (Ripple et al., 2014, 2016). Second, these species are impacted by conflict with humans, either through direct mortality, or via human hunting of their prey base (Pitman et al., 2015; Wallach et al., 2015) . Third, these species are officially recognized by institutions and have important cultural value to local communities in Colombia, and are therefore included in national and regional management plans (Castaño-Urbe et al., 2013; Suárez-Castro & Ramírez-Chaves, 2015) “

Page 3 L47–50: Recently there was a data paper that presented a combined database of all neotropical mammals. I wonder if the authors used this information as well and/or were part of that exercise.

We conducted a thorough review for each species using Web of Science, including all the existing information until 2018. We reviewed at that moment the information of two global databases (IUCN and Pantheria) and find no significant differences that substantially our results. We acknowledge the existence of one data paper from 2020 regarding Neotropical Carnivores (Nagy-Reis et al., 2020). We did not use this information as it was released after our analysis, but we did include the data from the universities and institutions from the Caribbean region that participate in this paper. We did use a more extensive dataset from different sources working with nearly 4000 occurrences just for our study region.

Page 4 L22–27: I really appreciate that the authors are being forthright about this. This would entail some subjectivity but is justified given the general scarcity of information from a lot of biodiversity-rich countries that do not have a long history of systematic monitoring of wildlife and habitats.

We thank the reviewer for this comment – especially regarding data scarcity; we have tried to be as clear and transparent as possible with data use within the paper and the supplemental material which allows for a traceable approach for future reproducibility. All methods that we used for data summary for each species can be found in the supporting information.

Page 5 L54–57: Please consider presenting a table in the main MS, with a list of species against all habitats considered, and the corresponding resistance value used for each species.

Considering that each species has a specific table regarding habitat identified, source of the information, and a species-specific landscape resistance, we created a detailed description of the information used in each modeling step within the supplemental material. We would prefer to retain the table in the supplemental material given the size and amount of information. We included an extra sentence explaining the contents of the supplemental material. We also developed a new supplemental material with all species information regarding habitats considered. We included (page 9 Line 33) our comment “However, to maximize reproducibility of this analysis, we have reported each of the variables used for each species in the supplemental material including the information used to select habitat types, home range, and dispersal distance analysis, detailed information of covariables used for niche modelling, spatial representations of niche and habitat patches, spatial representation of species specific resistances, and spatial representation of priority patches (Electronic supplementary material S 1, S 2, S3). ”.

1. The analysis did not involve “weighting” of individual species, i.e., endangered or rare species, or those that were more vulnerable to demographic effects of metapopulation level disruptions were not given higher

weights while using the combined connectivity areas. The authors should deliberate briefly on if and how this would alter their results; or, if similar studies are carried out using a wider set of species, how would incorporating these weights improve the assessments. There's quite a bit of spatial prioritization literature that the authors can look into for this.

We did not include weighting, as we tried to show the individual effect on the species of the group, and apply an approach that allows evaluation of the effect of different landscape conservation for each species. Species weighting prioritization is useful when making one final decision, or using a spatial prioritization algorithm, but can mislead conservation outputs for certain species that do not align with the distribution of the species with higher weights, as has been shown when using surrogate species (Devictor et al., 2010; Grantham et al., 2010; Levin et al., 2010; Whitehead et al., 2014).

We present Figure 5 and 6 for species-specific results which can be used in conservation decision making without using a spatially weighted prioritization. Prioritization is a further step required by the regional and local stakeholders; we hope this paper gives them sufficient information to lead such prioritizations and decisions.

We hope that species-specific results can be more informative than a weighted prioritization scheme but acknowledge that this is an important next step. We included a specific paragraph (page 12 Line 22) "Our analysis did not provide a multi-species weighted prioritization, as our objective was not to provide a single solution for multiple species. The main objective of this analysis was to provide evidence of the potential conservation outcomes of different landscape conservation decisions on several species with different spatial requirements. This approach enables quantification of the differential impacts of different approaches over a diverse but phylogenetically related species group. Nevertheless, we did find that priority patches overlapped across many species. These spatial results could be included as a conservation target in a spatial prioritization analysis to further decide where to create new PAs or other Effective Area-based Conservation Measures (Beger et al., 2015; Correa Ayram et al., 2015; Santini et al., 2016). They can also be used in surrogate species approaches when stakeholders require species-specific management strategies, as is the case with several endangered species (de la Torre et al., 2017; Thornton et al., 2016)."

A key component that I felt was missing in the Discussion was some comparison of economic or social costs of declaring PAs versus implementing PES. I fully understand that it was not the main focus of the analysis. Since the authors propose that both approaches may be useful, but PAs would definitely benefit some species better, it would be worthwhile to present some information on the financial costs and/or social costs of both these initiatives.

We agree that this is a bit beyond the scope of our paper. Unfortunately, there is not a general understanding of the comparative financial and social cost between these two conservation measures. PES, despite its growth in recent years, has a wide array of implementation approaches that impedes a realistic estimate of a single financial "cost" (Salzman et al., 2018). In Colombia alone, PES has been present since 2002 and currently more than 15 different programs exist. Those programs started without a clear legal framework (the legislation regarding PES was published in 2017) or a methodological approach and were stimulated by both local governments and private sector with unique approaches, which created information dispersion. A rough cost comparison can be done between the cost of PA maintenance (3.71-5.75 dollars/ ha) and the cost of PES payments (53-159

dollars/ ha) to a private owner (Bovarnick1 et al., 2010; Moros et al., 2020), but this does reflect social costs, which can also vary between countries, and implementation approaches (Grima et al., 2016).

Nevertheless, we have revised the discussion to reflect current discussions regarding costs and effectiveness of PES versus PA. We have included a specific sentence (page 11 line 44) “Overall, it is likely that the designation of new protected areas could have a greater impact on halting further habitat loss and warrant short-term conservation for most species; large, contiguous reserves are also more resilient to direct anthropogenic pressures such as bushmeat hunting (Barlow et al., 2016). However, such reserves will require greater long-term economic support from government funds, which is unlikely in Colombia where funding is low (Bovarnick1 et al., 2010). PES implementation often results in conservation of smaller forest fragments, that are likely to be more vulnerable to these harder-to-monitor drivers of biodiversity loss. On the other hand, an efficient PES scheme should both conserve and generate revenue to local communities which can have greater social impact and lower direct financial cost to the government over the long term; this assumes that PES is designed with clear transaction rules that allow continued contribution to local livelihoods (Grima et al., 2016). Unfortunately, to date, quantitative comparisons of economic and social costs versus conservation benefits of PES versus PAs has not yet occurred in Colombia, as it is a complex calculation that must take into account the broad array of PES schemes – most of which have not been evaluated for social and economic impacts (Salzman et al., 2018).”

Figures: The captions need not start with “Boxplot showing...”. You can directly describe the figure without saying it’s a boxplot in every instance.

We made the changes in figure 4 and 6.

Finally, I suggest the authors thoroughly proofread the manuscript for typographical and grammatical errors.

We acknowledge the comment and carried a thorough revision of the entire MS.

Response to Reviewer number 2:

Review: "Connectivity conservation at the crossroads: protected areas vs payments for ecosystem services in conserving connectivity for Colombian carnivores". Royal Society Open Science. The authors should firstly be applauded for their efforts to bring together such an ambitious study, particularly given the inevitable restrictions on data availability for many of the species they focus on. The volume of work here is impressive, and I am sure there are numerous novel insights emerging from this collation.

However, the authors ask a lot of the reader to trust the outputs. Whilst each step was thorough in many regards, I was frequently frustrated that vital details needed to follow the logic were often missing. Each of the datasets are treated to a number of manipulations before they are combined into the final analysis. Many of these treat each species or each dataset as equivalent, which is not necessarily the case. A more nuanced approach to data treatment prior to modelling, perhaps allied to simpler modelling approaches, particularly where sample sizes are small, would make this all the stronger.

Considering the lack of information and overall data scarcity for most of the species and the study in general, our modelling procedures are only meant to standardize the data available for all species, so that comparisons are sound and valid. However, we also acknowledge that the diverse modeling and methodological steps can be challenging to follow. On the advice of this reviewer we have now more clearly explained the methodological process in the MS and the detailed supplemental material. We added details at the beginning of the methods section to shed light on some of the most critical aspects of the modeling. We now include paragraph in the Methods section explaining our approach and have moved Fig. 2 (a conceptual/methods figure) further up in the MS, (page 4 Line 4) "Our methodological approach uses occurrence data, environmental and land-cover information and existing ecological knowledge, within a framework that reduces spatial bias in occurrence data (Kramer-Schadt et al., 2013). This framework includes for each species: i) quantifying potential distributions (Muscarella et al., 2014), ii) identifying suitable habitat patches (based on national land cover cartography) (Zárrate-Charry et al., 2018a), and, iii) defining a connectivity network between suitable habitat patches (based on ecological information), allowing a species-specific prioritization approach (Baranyi et al., 2011; Correa Ayram et al., 2015). We then use the whole framework as an input for comparative landscape analysis to assess the potential effectiveness of PAs vs PES. [¡Error! No se encuentra el origen de la referencia.]

We developed this methodological framework to ensure a standardized approach for the identification of priority connectivity patches for multiple species in the context of data scarcity. This approach, and the resulting outputs, allows its use by regional and local stakeholders within existing landscape planning instruments and cartographic information (IDEAM et al., 2007; MADS, 2014). We used the same methods and followed the same previously published modelling and data management procedures for all species (Correa Ayram et al., 2015; McRae et al., 2008; Muscarella et al., 2014; Radosavljevic & Anderson, 2014), thus allowing for sound inter-species comparisons"

Where arbitrary cut-off values have been used, it would be helpful to show sensitivity analyses in subsequent models to understand the relative impacts of small changes to each derived parameter value.

We based modeling decisions on existing published methods, but they were not arbitrarily selected.

Below, we include a list of decisions that we made, their support (in the literature and data) and the location within the main text where they are justified. Here, we highlight only parts that were changed. We also moved the section of data sources to the end of the methods section. Also, each final attribute used is included in the supplemental material in detail for each species.

1. Potential distribution: We used only previously used methods of modeling to construct binary potential distributions of each species, the we acknowledge that different decisions were possible, but the sensitivity of results to such alterations to modelling approach has already been previously analyzed extensively in the literature.
 - a. Species occurrence thinning (page 5 line 45): “To identify clusters of occurrences, we conducted species-specific spatial thinning based on the distribution of the existing occurrence records and used a pairwise distance matrix between occurrence points to identify potential data clustering (Peterson et al., 2011), (Peterson et al., 2011), and created a species-specific background based on biogeographic units where the species has been reported (Anderson & Gonzalez, 2011). The species-specific thinning used and the numbers of occurrences removed by the spatial thinning is reported within the supplemental material.”
 - b. Environmental variable selection. “We evaluated the correlation among environmental variables used to train the models and removed variables that were highly correlated (i.e., a Spearman rank correlation > 0.8). We validated the statistical significance of all variables used in each model using a jack-knife analysis (Phillips et al., 2006).”
 - c. Model selection: “We estimated the optimal model complexity for each species as follows. We ran 30 Maxent models per species changing the regularization multiplier and the combinations of the used feature classes. We used six regularization multipliers (i.e., 0.5, 1, 1.5, 2, 2.5, 3) and five feature class options (i.e., linear, linear/quadratic, linear/quadratic/product, linear/quadratic/product/threshold, linear/quadratic/product/threshold/hinge) (Muscarella et al., 2014; Shcheglovitova & Anderson, 2013). The performance of each model was evaluated using a cross-validated masked checkboard subset of randomly selected occurrence points (Muscarella et al., 2014; Radosavljevic & Anderson, 2014). We used AIC to choose the best performing model for each species and retained only those with the Area Under the Receiver Operating Curve (AUC) values higher than 0.7. We developed all models using the ENMeval package for R (Muscarella et al., 2014).”
 - d. Binary output: “The final selected model was used to construct a binary output using the minimum training presence logistic threshold, which reduces omission errors (Liu et al., 2005). We then refined this binary output using the maximum and minimum elevational limits for each species (Hunter, 2011; Ocampo-Peñuela & Pimm, 2014; Suárez-Castro & Ramírez-Chaves, 2015; D. E. Wilson & Mittermier, 2009). The “potential distribution” or environmental niche is understood as the suitable abiotic environment of a species (Guisan et al., 2017).”
2. Habitat patches: This step presents two decisions, both of which are based on ecological needs for the species. We expect that results may be sensitive to cut points used to quantify habitat patches, but we argue that our ecologically informed approach is well justified. In particular, we identified only habitat patches that were larger than the home range of each species. Patches smaller than the individual home range can be important under particular conditions, but we strove for conservative results. We include the following comments within the paper.

- a. Land use land cover defines as habitat: “Within each identified potential distribution, we used regional land-cover data to identify all suitable habitat patches that had at least one of the cover types reported in the literature as being habitat for each species”
 - b. Habitat patch size: “We further filtered the final number of patches present in the region by removing those that were smaller than the individual median reported home-range of each species. We defined patches in this way to reflect the need for population connectivity (individuals must reside and survive in a patch while moving between sub-populations) (Di Minin et al., 2013; Saura et al., 2014). We acknowledge that some species may also use smaller patches as temporary steppingstones (Saura et al., 2014), so our analysis should be considered conservative in this regard.”
3. Connectivity areas: This is the only part of the paper where we acknowledge that a more subjective decision was necessary (Armenteras et al., 2015; Moilanen et al., 2009; Rondinini et al., 2006; K. a. Wilson et al., 2005). However, determining “how much habitat is enough” is still a highly debated question in conservation biology (Arroyo-Rodríguez et al., 2020), and we justified our choice in the literature. We decided to recommend a habitat amount for conservation of connectivity based on Di Minin and collaborators (2016). This approach has been used previously by the IUCN mammal specialist group. We could have used a different threshold based on IUCN red list status, or the importance of the species as surrogate, but that would only result in a different rather arbitrary decision. We include here an image showing the number of patches we used based on the 50% threshold.

Figure 1. Comparison between total number of patches and priority connectivity patches selected using a 50% threshold.

I agree entirely with the author's comments that issues of data scarcity, particularly in developing countries, should not prevent attempts to draw out important information that can assist with efforts to protect biodiversity. As I say, I can't help feeling that this paper would have been stronger if a more simple approach had been taken. This often appears somewhat overly complex, with no strong rationale for why this complexity is required. This this sadly casts doubt on the results, which is a pity, as I am sure there is are many value findings that need to be presented.

We acknowledge and thank the reviewer comments regarding data poor countries. Currently Colombia is facing deforestation and land-use changes in all fronts. The use of simpler analysis models can potentially provide guidelines for the creation or design of protected areas or zones where to implement PES. There are a couple of those approaches have been used in Colombia. An example of this is the system of protected areas in Colombia, which since 2010 has been using a systematic conservation planning process using ecoregions as an element of analysis, this being the only biodiversity criterion used.

Unfortunately, this approach does not align with the needs of the species, as has been evidenced at the national and international level. This makes it necessary to include criteria for species, populations and communities when conducting landscape planning.

There are also national and international approaches to define the distribution of species, such as the distribution maps of the IUCN or the models that are being developed by the country's biodiversity research institute (IAVH). Both approaches are simpler than our approach but have severe limitations. In the case of the IUCN distribution models, they over represent the coarse-resolution distribution of species which has the effect of both under- and overestimating species distributions (Fig. 2).

Figure 2. Maps of distribution of three species a) *H. yagouaroundi*, b) *G. vittata*, c) *L. pardalis*, showing the IUCN species distribution (red polygon), the occurrence records (yellow dots), and the distribution of connectivity patches (green polygons).

Other researchers have simplified conservation planning by identifying human-defined cover types within potential species distributions, but this often does not reflect species distributions or habitat preferences (Betts et al., 2014).

Further, prior to our work, there have been no efforts to quantify species connectivity in Colombia, and its used within protected areas design has been limited, but is currently gaining importance based on its inclusion within national and international requirements of protected areas guidelines. Thus, our analysis in addition to being more ecologically realistic and fills an important gap in knowledge with respect to connectivity.

Although our approach has several methodological complexities, its objective is to provide an spatial representations that allow explicit spatial representation of the species-specific needs to be efficiently included in planning tools. This has already been used by local authorities in Colombia, and is not unlike several other recent studies (Areiza, A., Corzo, G., Castillo, S., Matallana, C. y Correa Ayram, 2018; SINAP, 2019; Zárrate-Charry et al., 2018b) (<https://es.mongabay.com/2021/01/jaguar-sierra-nevada-de-santa-marta-conservacion-colombia/>).

I am still not entirely clear what the comparison between the PA and PES approaches actually entails. For example, what are the differences between the two that might lead to changes in mammalian carnivore occupancy and movement? Would there be suitable habitats for each species for instance and how would that come about? Given the aims of the paper, more detail on the distinction between PA and PES strategies is needed.

The distinction between these two landscape conservation measures varies depending on country, economic strategy, and environmental services that can be included according to local legislation (Goldman-Benner et al., 2012; Ruckelshaus et al., 2013). We included a more detailed description of the differences and potential outcomes in the introduction, following a similar comment of another reviewer. (page 2 Line 16) "PAs have been the primary biodiversity conservation strategy worldwide, with strong positive effects on forest and ecosystem conservation (Le Saout et al., 2013; Morales-Hidalgo et al., 2015). The total area of PAs has increased in tropical countries, from 12% in 1990 to 26.3% in 2015, an increase that is three times greater than in boreal and temperate zones (Morales-Hidalgo et al., 2015). On the other hand, in the last two decades, PES has been presented as an economic incentive for developing countries to mitigate climate change by reducing deforestation and forest degradation, while protecting high biodiversity areas and improving local livelihoods (Daily & Matson, 2008; Venter et al., 2013). The inclusion of poverty alleviation considerations within PES schemes and its environmental and economic potential impact could shift legislative and management efforts from PAs to PES, prioritizing social poverty alleviation above environmental goals (Chan et al., 2017; Miller et al., 2011). In Colombia, such pattern is still not clear, and even when the legal framework was already developed, clear protocols on how or where to implement PES are vague and diverse, and its implementation depends on the funding source, implementation agency and regional factors (Grima et al., 2016; Moros et al., 2020; Redacción Comercial del Espectador, 2017). Even when it is not clear yet if PAs potential creation is decreasing with the increasing on PES schemes implementation, there is an increasing trend for promoting PES conservation actions as a reflection of a widespread narrative shifting from strict protection to other conservation strategies (Santamaría et al., 2018; SINAP, 2019), even when its conservation success in terms of forest conversion is lower (Areiza, A., Corzo, G., Castillo, S., Matallana, C. y Correa Ayram, 2018)".

We have also included additional information differences between PES and PAs in the Discussion (page 11 Line 29) “Overall, it is likely that the designation of new protected areas could have a greater impact on halting further habitat loss and warrant short-term conservation for most species; large, contiguous reserves are also more resilient to direct anthropogenic pressures such as bushmeat hunting (Barlow et al., 2016). however, such reserves will require greater long-term economic support from government funds, which is unlikely in Colombia where funding is low (Bovarnick et al., 2010). PES implementation often results in conservation of smaller forest fragments, that are likely to be more vulnerable to these harder-to-monitor drivers of biodiversity loss. On the other hand, an efficient PES scheme should both conserve and generate revenue to local communities which can have greater social impact and lower direct financial cost to the government over the long term; this assumes that PES is designed with clear transaction rules that allow continued contribution to local livelihoods (Grima et al., 2016). Unfortunately, to date, quantitative comparisons of economic and social costs versus conservation benefits of PES versus PAs has not yet occurred in Colombia, as it is a complex calculation that must take into account the broad array of PES schemes – most of which have not been evaluated for social and economic impacts (Salzman et al., 2018) ”

Finally regarding movement, we include a paragraph in the discussion focused on why the PA approach is more likely to ensure better movement between sub-populations than the PES as is currently implemented (Page 11 line 36) “We focused our analysis on priority connectivity patches, ensuring that patches reflect population requirements, meaning they are sufficiently large for single individual territory while moving between sub-populations (Saura et al., 2014). In this respect, protected areas are more likely to ensure movement between patches as are larger than the individual home range of all species in our study ($\bar{x} = 33.8 \text{ km}^2 \pm 64.03 \text{ km}^2 \text{ SD}$). In contrast lands selected for PES tended to be smaller than the home ranges of many species we considered (mean forest size = 15 hectares (APC-Colombia & Fondo Acción, 2016))”

One of the main global threats to mammalian carnivores is conflict-driven persecution. It would be interesting to see a little more consideration of this in light of the differences between PAs and PES. PAs essentially seek to separate wildlife from people, whereas PES potentially seeks a more integrated model. It is difficult to argue livestock depredation or risk to people’s lives as being an ecosystem service worthy of payment, so the PES approach is immediately problematic for this taxon. It is arguable that any fully functioning restored or protected natural or semi-natural ecosystem will include predators as an essential part of its trophic structure, e.g. limited populations of potentially problematic prey or meso-predators.

This is an excellent point. We have now added a few sentences to the discussion highlighting that PES schemes – because they tend to integrate humans with nature – tend to increase the risk of human-wildlife conflict. These should be included when decision making is developed but it is not the focus of our analysis. We included a paragraph within conclusion (page 14 line 24). “Other aspects of carnivore conservation, regarding conflict driven persecution should also be analyzed when making landscape conservation decisions Landscape conservation directly allows movement of carnivores (Di Minin et al., 2016; Mateo-Tomás et al., 2012; J. W. Wilson et al., 2019) which can generate conflict, depending on the landscape use and amount of wildlife hunting within these areas, which will affect species resources and will direct them and increase chances of predation toward domestic animals (Inskip & Zimmermann, 2009; Ripple et al., 2014). The current proposal seeks to analyze alternatives for conservation of basic population parameters necessary of the species connectivity, but when implemented other dimensions of carnivore conservation should be included ”

In this context, what are the goals of PES (i.e. what ecosystem services are being paid for) and what are the plausible consequences for both people and wildlife? I offer my more detailed thoughts below. Line numbers indicated are approximate, as they don't quite align with the text in the draft I received.

We have now included more detailed information regarding the implementation of PES in Colombia, and the differences within Latin-American countries.

Page 3; Lines 14-20: Are PES and PA actually being presented as alternatives? As you point out, they are focused on different things, so is there evidence of PA loss as a result of enhanced PES, for example?

There is no direct evidence that a focus on PES is reducing PA creation, but certainly regional and local efforts are shifting toward PES and private reserves (Santamaría et al., 2018; SINAP, 2019). Since 2010 we have created just three new protected areas (56 to 59) while private reserves have increased from 256 to 326 (Areiza, A., Corzo, G., Castillo, S., Matallana, C. y Correa Ayram, 2018).

Second, the National Protected Area System is massively underfunded. The focus recently has been on maintaining existing protected areas. For instance, one PA used in our analysis has been identified by the Colombian government (10 years ago!), but remains unestablished (Serranía de San Lucas, Serranía de Perijá). Currently there are other regional or local initiatives aimed at protecting some of it through alternative conservation strategies that are aligned with categories IV, V and VI of IUCN (e.g., PES, communal reserves, peasant reserve areas, private reserves). We included a comment regarding this reality within the introduction. (Page 2, line 24) "In Colombia, such pattern is still not clear, and even when the legal framework was already developed, clear protocols on how or where to implement PES are vague and diverse, and its implementation depends on the funding source, implementation agency and regional factors (Grima et al., 2016; Moros et al., 2020; Redacción Comercial del Espectador, 2017). Even when it is not clear yet if PAs potential creation is decreasing with the increasing on PES schemes implementation, there is an increasing trend for promoting PES conservation actions as a reflection of a widespread narrative shifting from strict protection to other conservation strategies (Santamaría et al., 2018; SINAP, 2019), even when its conservation success in terms of forest conversion is lower (Areiza, A., Corzo, G., Castillo, S., Matallana, C. y Correa Ayram, 2018)."

Page 3; Lines 53-55: What is the distribution of PES across the region? Is the 130km2 contiguous or scattered? Why are these PES there? Are PES measures put in place in response to EIA requirements or some other intervention?

We provide details in the MS to clarify this point.(Page 3 line 9) "PES in Colombia are still not widespread, and the most extensive initiative so far, based on payment for natural ecosystems conservation, covers around 130 km²; these areas are primarily located within small private farms (mean forest size = 15 hectares (APC-Colombia & Fondo Acción, 2016)), that are scattered and opportunistically dispersed across multiple landscapes; neither distribution/location or size have been a limitation for postulation (Moros et al., 2020; Redacción Comercial del Espectador, 2017). In fact, location of selected farms is not systematic and is guided mainly by state-specific interests or by a voluntary commitment by landowners to the program (Redacción Comercial del Espectador, 2017)"

Page 4; Line 8: What is being proposed in the "proposed PES schemes"? It's not clear what these are intended to do, nor whether they are reactive or proactive.

There is not a clear protocol or path to evaluate if they are reactive or proactive, but the legislation has already defined some services that can be focus of payment (water provision, carbon sequestration, cultural areas). The implementation of such payments is not clear, not centralized and still requires definition. We included the services that are included within the legislation for clarification, but other details are still missing within the national implementation. We included a clarification regarding the current state of PES implementation in the text (Page 2 line 24) “In Colombia, such pattern is still not clear, and even when the legal framework was already developed, clear protocols on how or where to implement PES are vague and diverse, and its implementation depends on the funding source, implementation agency and regional reality (Grima et al., 2016; Moros et al., 2020; Redacción Comercial del Espectador, 2017).”

Page 4; Lines 12-13: “discussion on the expansion of both” is a little vague. It would be helpful to see more details about what is being proposed and how these two approaches are being weighed against each other.

This is a new and ongoing discussion, and we do not yet have a clear understanding on how these approaches are going to be weighed against each other. The legal framework is being developed and regional and national documents are showing diverse narratives but there is not a pattern yet. Indeed, this is why we attempted to tackle the question of relative conservation benefits of PES versus PAs in this MS. Nevertheless, we have expanded our explanation in this revised MS.

Page 5; Lines 2-8: this is potentially a good way to incorporate a wide range of data from different sources and methodologies. However, you do not appear to have accounted for possible errors or biases between studies/methods, particularly given the differences in the number of report/papers for each species. Your approach of taking estimates and using them in subsequent calculations has the potential to further compound these errors. Would it have been possible to explore these errors a little more before settling on parameter values? For example, a limited radio tracking study of a small number of a large species should not be treated as equivalent to a larger study using GPS technology. Similarly, telemetry of all types is not necessarily equivalent for species in different habitats (open vs closed habitats for example).

We agree with this reviewer comment. This is one of the main issues when using data that come from diverse sources, regions, and research studies. Unfortunately, discarding studies with low sample size would have meant having no estimates for many species. We included a paragraph in the discussion to address this issue (page 14, line 9) “We developed a spatial analysis that allows the identification of conservation needs in a species-specific framework for a data poor country. We also created an output that can be aligned with national and regional geographic information, as it follows the national land-use and land-cover typology (Feranec et al., 2007; Zárrate-Charry et al., 2018b). Although we acknowledge our approach requires validation, and extensive research to reduce uncertainty in the ecological information, still it can be considered a novel approach to include biodiversity information in a landscape planning that has historically used only coarse ecoregion data to design protected areas”

Page 5; Lines 10-11: by “scenopoetic, non-dynamically linked environmental variables”, do you mean ‘independent’ or is there some other relationship between variables? Either way, how did you determine this? The tables in ESM1 (e.g. Table 4) do not provide sufficient information about these variables, including the selection and contribution statistics used.

Scenopoetic variables are defined as the variables that can affect the fitness and/or limit the distribution of a species, but are not affected or transformed by the species in any way (Peterson et al., 2011; Soberón, 2007). This definition comes from the definition of ecological niche of Hutchinson and highlights the difference between Grinnellian and Eltonian factors. The most common scenopoetic variables are the climatic variables. We have revised the text to clarify "For ecological niche models, we used only scenopoetic, non-dynamically linked environmental variables (Peterson et al., 2011). We used 19 climatic variables obtained from WorldClim version 2 (Fick & Hijmans, 2017)."

We performed a correlation analysis to select from the 19 variables, for use in BIOCLIM, that were not correlated within the model calibration area. We now present the method as follows "Before fitting the model, for each species, we evaluated the correlation among environmental variables used to train the model and removed variables that were highly correlated (i.e., a Spearman rank correlation > 0.8)."

Page 5; Lines 22-24: Given the inevitable discrepancies between difference land cover or habitat definitions, how did you make these "subjective decisions"? It's good that you have provided a comprehensive comparison of habitats for each species (e.g. ESM1 Table 2), but how were these then related to the Colombian national maps from which you defined the patches?

We now include one extra table within the supplemental materials that provides information on species' habitat types. We used a direct translation of the cover within each species, and if the cover type was too general we aligned it with the more specified land cover of the national cover (Forest = Dense forest, Dry Forest). We have the detailed tabulation of each coverage in comparison to the national coverage and have included it within this response but believe that level of detail is not needed. We included as a new as supplemental information (Electronic supplementary material S 3).

Page 5; Lines 46-52: while these methods might overcome some of the limitations of the often small and spatially limited species occurrence data, I think it's asking a lot in some cases. For species with the fewest points, why bother with MaxEnt at all? This is arguably making the analysis appear less plausible, given the extent to which you are pushing these limited data. Would a simple habitat definition based on published data applied to the land cover maps have sufficed? From some AUC plots (e.g. ESM1 fig. 1) there seems to be little convergence.

As we noted above, we argue that using only published data of just cover-type associations or raw Maxent models can under- or over-represent species distributions. We have now more clearly emphasized this point in the main text. We included within the occlusion an statement regarding this point (page 14 line 15) "Species-specific analysis has substantial benefits in terms of quantifying landscape-scale anthropogenic effects on biodiversity (Betts et al., 2014; Hadley & Betts, 2016), something which has been historically overlooked in developing megadiverse countries like Colombia. Previous global analysis has used less detailed and habitat definitions, which we think has been detrimental to conservation of habitat of species in regional landscapes as the Colombian Caribbean region, and has been seen with some of the more known and charismatic species (de la Torre et al., 2017; Zárrate-Charry et al., 2018b) The use of simplified existing information from global analysis can both over or sub present the actual distribution of an species, and can overlook endangered subpopulations (de la Torre et al., 2017; Zárrate-Charry et al., 2018b)"

Page 6; Line 10: is the "minimum training presence logistic threshold" given as "MTLPT" in ESM1 (e.g. table 5)? If so, is this the threshold above which habitats were considered suitable? This needs to be clearer as for

some species the MTLPT is very low (e.g. 0.07 in ESM1 Table 10). Given that this threshold appears only to use training data, doesn't the minimum risk over-estimating the extent of suitable habitat? What was the training/test split?

We used MTLP to conservatively minimize omission errors, and then refined the output based on the elevational limits of each species. Even though in some species MLPT can be low, this was only the first step in defining species' habitat. Next, we selected only cover types associated with suitable habitat for each species and then filtered these areas by species-specific home range sizes to define patches. We have revised the Methods to clarify (page 5 line 24): "We choose this set of rules to ensure that each identified suitable habitat patch had the both the cover type *and* climatic variables to support each species. Finally, these patches had to be of sufficient size to meet individual species' ecological requirements (Electronic Supplemental Material 3)."

Page 6; Lines 53-56: It is not clear how you derived the resistance layers for each species. From the text it appears that you used one resistance model for all species, based on the review paper by Zeller et al (2012), but I'm not sure if this is correct. Why did you not use an inverse function of the habitat suitability for resistance, on the basis that each species will move through its preferred habitats more readily than through its least preferred habitats? This would seem to define cost-pathways for each species better than an arbitrary definition based on human structures etc. These variables should have been included in species habitat suitability models, so their relative impact on each species could be explored more fully. Where these relationships are poorly understood, then a range of plausible values could be employed to look at their likely impacts, relative to other determinants of species occurrence.

We did include areas outside the identified potential distribution as a restriction to movement, as can be seen in the supplemental material 2. We did this separately for each species. We did not use the continuous output of the Maxent model, but as a binary output where areas within the niche generate no restriction and areas outside the niche generate a maximum value restriction. We then developed a weighed overlay between the binary model and the sum of all other landscape variables that limit movement (distance to roads, distance to cities, human density, slope, aboveground biomass, land coverage) based on Zeller et al (2012). We have now included a clarification in the main text. (Page 7 line 2) "To specify the least-cost-path distances we used resistance layers developed from previously reported variables that are known to affect movement of terrestrial carnivores [79]. We also assumed that areas outside of a species' potential distribution impeded movement. We imposed a restriction on movement for areas that fell outside of our quantitative definitions of "habitat" for each species; for simplicity we included habitat as a dichotomous rather than continuous (probability-scale) variable".

Page 7: Line 24: what were the units of the three indices summed to give a connectivity rank? Were they on equal scales to prevent any index overwhelming the calculation?

Yes we transformed each index value to ensure they have equal scales. We now highlight this more clearly in the text (page 7 line 39).

Page 7: Lines 31-32: I don't follow this final sentence of the paragraph. What is 'greater habitat connectivity importance' Is it all areas/patches within the top 50% of the ranking? As you say, this is an arbitrary cut-off, so it would be helpful to see an sensitivity analysis. For example, what is the change in patch number / area if this threshold is altered?

After we ranked patches, we selected each patches one by one until 50% of the total habitat area for each species was included. This allowed us to ensure that we are both selecting the patches of greater importance for connectivity and that we meet the proposed minimum area required for the conservation of carnivore species as presented by Di Minin and collaborators (2016). We included a clarification (Page 7 line 40) “We selected patches in rank order of their connectivity importance score until we reached a total of 50% habitat conserved. This is the threshold established by Di Mini et al (2016) for effective conservation of carnivores”. We chose not to do sensitivity analysis around this 50% value because we did not expect the main finding of our analysis – the differences between PES and PAs – to be influenced by this threshold.

Page 7: Lines 47-50: what is the legislation that leads you to think that these areas ‘could be’ PES options? This is somewhat vague, so please provide a reference.

The national legislation of environmental services is quite new in Colombia (Decreto Ley 870 de 2017). It was released in 2017 and the implementation has been slow and protocols and rules for implementations are just beginning. Currently the legal framework has identified four services that can be included in PES mechanisms (1. regulation and water quality services, 2. Reduction and storage of greenhouse gases, 3. Biodiversity conservation, 4. cultural, spiritual, and recreational environmental services). For the cultural and biodiversity services there is not currently any detailed approach and there are just some pilots at the national level. Regulation and water quality services (Decreto 953 de 2013), and Reduction and storage of greenhouse gases (Colombian Low Carbon Development Strategy (ECDBC)), have pilots, and other legislation that supports them, which supports our use of such criteria in the selection of potential PES areas.

We also used a national analysis that used the information of potential environmental services location that was constructed using the cartography of the national he Institute of Hydrology, Meteorology and Environmental Studies (IDEAM), which is the main research institute of the country regarding the distribution of the identified services (Rodríguez et al., 2015). We included clarification within the text. (page 7 line 37). “As areas of potential PES implementation, we used the environmental services spatial prioritization constructed by Rodríguez and collaborators (2015), which uses environmental services currently supported by Colombian national legislation (Decreto Ley 870 de 2017).”

Page 7; Lines 51-53: on what basis do you make the assumption that PES would be implemented over an area of 52,683km² (27% of the study region), particularly in light of your introduction, that states the most extensive area of PES in Colombia is 130km² ?

We have now provided further justification for the assumption that PES areas are likely to expand substantially in future decades. (page 12, line 11). “We acknowledge that an extensive implementation of PES may be unlikely in the short term, but there is an ongoing discussion about the global distribution of PES schemes and payments. Therefore, the attribution of 27% of the area of Colombia under PES schemes seems feasible [101,102]. Our main objective was to show that the distribution of areas that have high potential for provision of ecosystem services could align with areas of importance for carnivore conservation”.

We also made the same clarification within the methods (page 8, line 29). “We acknowledge that the area defined within the PES implementation is close to 27% of the study region, which is an ambitious

goal for PES implementation, but here we are trying to identify the overlap between distribution of areas that have high potential for provision of ecosystem services and areas of importance for carnivore conservation.”

Pages 7 Line 56 – Page 8 Line 15: This section seems pivotal to the entire paper, in describing how PA and PES approaches were compared. In common with the introduction, I am not clear to what extent PES is actually comparable with PAs in the context presented. This clearly causes problems for the paper, so I would urge the authors to make this clearer. What does ‘we then implemented PES’ in these 1km² pixels actually mean? Was there a change in land use or land cover as a consequence, for example? What is the change that might be expected to somehow alter movement and occupancy of mammalian carnivores in these areas? How do these two proposed PAs relate to the study area in terms of comparable habitats, carnivore assemblages etc?

We included several comments through the document to try to explain in more detail the rationale of our approach based on the current discussion of PES implementation.

We highlight here some of the previous inclusion to highlight how this comment has been answered.

What does ‘we then implemented PES’ in these 1km² pixels actually mean?

We have already established that the area defined within PES is smaller than a PA, which creates scattered small areas that have a mean size of 0.15 km². We used a 1 km² area for two reasons 1) this is the scale of our geographic information, 2) we tried to ensure a comparison was on an even footing. This is highlighted both in the introduction, methods and discussion.

Was there a change in land use or land cover as a consequence, for example?

We currently do not know as there has not been a clear analysis of the implementation within the country, this has been highlighted both in the introduction and in the discussion and conclusions.

What is the change that might be expected to somehow alter movement and occupancy of mammalian carnivores in these areas?

We currently do not know the impact or effect of this distribution of landscape conservation decisions, but we acknowledge some potential gains and also problems based on your previous comments and include them in the discussion and conclusion sections.

How do these two proposed PAs relate to the study area in terms of comparable habitats, carnivore assemblages etc?

They are both located within the same national ecoregion (Region Caribe), but represent quite different ecosystems, that's why they are both selected within the systematic conservation planning that Colombia has followed the last 10 years (Andrade Pérez & Corzo Mora, 2011). In terms of carnivore assemblages both areas and the whole Caribbean region is quite similar, and the inference regarding species and comparison is the same, as can be seen in the distribution of habitat of the species in figure 3.

Page 8; Lines 42-43: If habitat suitability models did not include the human elements included in the resistance models, then perhaps it is not surprising to find habitats areas outside Pas. If this is intended to be the 'idealized' rather than the 'realized' habitat use of these species, then this needs to be clearer.

There is an important distinction between the (1) potential distribution of the species (the Maxent models including scenopoetic variables), (2) the representation of the suitable habitat patches (result of our identification of habitat patches following a species-specific [literature based] approach), and (3) the effect of human variables in the restriction of movement of the species. Thanks for encouraging us to make this clearer.

Please see our response to your request for more details on our modeling methods above. We hope this details on this three-step approach help clarify.

Page 8; Line 49-56: How was data from Global Forest Watch used within the connectivity analysis? I can't see any reference to it in the Methods section and figure 4 appears to show areas of loss.

Thank you for the comment. We did not use the GFW information within the analysis, we only used these data to show how much of the current forest within habitat patches has been lost based on the data of GFW. The point was to show the urgency of the protection of these areas.

Page 9; Line 32 - Are 'Pes' something different to Pas or should this be 'PES'?

We corrected this typo.

Page 10; Line 7-9: This statement needs support, particularly describing more fully the PES scenario used. PES can take many different forms, so further detail is required on how representative or realistic is this PES depiction.

We included a detailed description both in methods and discussion based on a previous comment of the reviewer.

ESM1 Page 440 Table 30: 200 occurrences minus 29 and 2 equals 169. Why only 122 entered into model. Also, the number of parameters (52) seems quite high for the amount of data used in the model.

Thank you very much for your detailed and constructive review. Indeed, there was an error in the species table related to Information regarding the Ecological Niche Model selected. We now include the number of occurrences that were filtered by the spatial thinning and the size in kilometers of the grid; the correct numbers are now reported in the supplemental material.

Bibliography

- Anderson, R. P., & Gonzalez, I. (2011). Species-specific tuning increases robustness to sampling bias in models of species distributions: An implementation with Maxent. *Ecological Modelling*, 222(15), 2796–2811. <https://doi.org/10.1016/j.ecolmodel.2011.04.011>
- Andrade Pérez, G. I., & Corzo Mora, G. A. (2011). *¿Qué y Dónde Conservar?* (First edit). Parques Nacionales Naturales de Colombia.
- APC-Colombia, & Fondo Acción. (2016). *BanCO2 caso de estudio*. https://www.apccolombia.gov.co/sites/default/files/archivos_usuario/casos/06-cartilla_-_banco2.pdf
- Areiza, A., Corzo, G., Castillo, S., Matallana, C. y Correa Ayram, C. A. (2018). Áreas protegidas regionales y reservas privadas: las protagonistas de las últimas décadas. In L. A. Moreno, G. I. Andrade, & M. F. Gómez (Eds.), *Biodiversidad 2018. Estado y tendencias de la biodiversidad continental de Colombia*. Instituto de Investigación de Recursos Biológicos Alexander von Humboldt.
- Armenteras, D., Rodríguez, N., & Retana, J. (2015). National and regional relationships of carbon storage and tropical biodiversity. *Biological Conservation*, 192, 378–386. <https://doi.org/10.1016/j.biocon.2015.10.014>
- Arroyo-Rodríguez, V., Fahrig, L., Tabarelli, M., Watling, J. I., Tischendorf, L., Benchimol, M., Cazetta, E., Faria, D., Leal, I. R., Melo, F. P. L., Morante-Filho, J. C., Santos, B. A., Arasa-Gisbert, R., Arce-Peña, N., Cervantes-López, M. J., Cudney-Valenzuela, S., Galán-Acedo, C., San-José, M., Vieira, I. C. G., ... Tschardtke, T. (2020). Designing optimal human-modified landscapes for forest biodiversity conservation. In *Ecology Letters* (Vol. 23, Issue 9, pp. 1404–1420). Blackwell Publishing Ltd. <https://doi.org/10.1111/ele.13535>
- Baranyi, G., Saura, S., Podani, J., & Jordán, F. (2011). Contribution of habitat patches to network connectivity: Redundancy and uniqueness of topological indices. *Ecological Indicators*, 11(5), 1301–1310. <https://doi.org/10.1016/j.ecolind.2011.02.003>
- Barlow, J., Lennox, G. D., Ferreira, J., Berenguer, E., Lees, A. C., Nally, R. Mac, Thomson, J. R., Ferraz, S. F. D. B., Louzada, J., Oliveira, V. H. F., Parry, L., Ribeiro De Castro Solar, R., Vieira, I. C. G., Aragaõ, L. E. O. C., Begotti, R. A., Braga, R. F., Cardoso, T. M., Jr, R. C. D. O., Souza, C. M., ... Gardner, T. A. (2016). Anthropogenic disturbance in tropical forests can double biodiversity loss from deforestation. *Nature*, 535(7610), 144–147. <https://doi.org/10.1038/nature18326>
- Beger, M., McGowan, J., Treml, E. A., Green, A. L., White, A. T., Wolff, N. H., Klein, C. J., Mumby, P. J., & Possingham, H. P. (2015). Integrating regional conservation priorities for multiple objectives into national policy. *Nature Communications*, 6, 8208. <https://doi.org/10.1038/ncomms9208>
- Betts, M. G., Fahrig, L., Hadley, A. S., Halstead, K. E., Bowman, J., Robinson, W. D., Wiens, J. a., & Lindenmayer, D. B. (2014). A species-centered approach for uncovering generalities in organism responses to habitat loss and fragmentation. *Ecography*, 37(6), 517–527. <https://doi.org/10.1111/ecog.00740>
- Bovarnick1, A., Fernandez-Baca, J., Galindo, J., & Negret, H. (2010). *Financial Sustainability of Protected Areas in Latin America and the Caribbean: Investment Policy Guidance* (UNDP (ed.)). United Nations Development Programme, The Nature Conservancy.
- Castaño-Uribe, C., González-Maya, J. F., Zárrate-Charry, D. A., Ange-Jaramillo, C., & Vela-Vargas., I. M. (2013). *Plan de Conservación de Felinos del Caribe colombiano: Los felinos y su papel en la planificación regional integral basada en especies clave*. Fundación Herencia Ambiental Caribe, ProCAT Colombia, The Sierra to Sea Institute.

- Chan, K. M. A., Anderson, E., Chapman, M., Jespersen, K., & Olmsted, P. (2017). Payments for Ecosystem Services: Rife With Problems and Potential—For Transformation Towards Sustainability. *Ecological Economics*, 140, 110–122. <https://doi.org/10.1016/j.ecolecon.2017.04.029>
- Correa Ayram, C. a., Mendoza, M. E., Etter, A., & Salicrup, D. R. P. (2015). Habitat connectivity in biodiversity conservation: A review of recent studies and applications. *Progress in Physical Geography*, 40(April), 1–32. <https://doi.org/10.1177/0309133315598713>
- Daily, G. C., & Matson, P. a. (2008). Ecosystem services: from theory to implementation. *Proceedings of the National Academy of Sciences of the United States of America*, 105(28), 9455–9456. <https://doi.org/10.1073/pnas.0804960105>
- de la Torre, J. A., González-Maya, J. F., Zarza, H., Ceballos, G., & Medellín, R. A. (2017). The jaguar's spots are darker than they appear: assessing the global conservation status of the jaguar *Panthera onca*. *Oryx*, 1–16. <https://doi.org/10.1017/S0030605316001046>
- Devictor, V., Mouillot, D., Meynard, C., Jiguet, F., Thuiller, W., & Mouquet, N. (2010). Spatial mismatch and congruence between taxonomic, phylogenetic and functional diversity: the need for integrative conservation strategies in a changing world. *Ecology Letters*, 13(8), 1030–1040. <https://doi.org/10.1111/j.1461-0248.2010.01493.x>
- Di Minin, E., Hunter, L. T. B., Balme, G. a, Smith, R. J., Goodman, P. S., & Slotow, R. (2013). Creating larger and better connected protected areas enhances the persistence of big game species in the maputaland-pondoland-albany biodiversity hotspot. *PloS One*, 8(8), e71788. <https://doi.org/10.1371/journal.pone.0071788>
- Di Minin, E., Slotow, R., Hunter, L. T. B., Montesino Pouzols, F., Toivonen, T., Verburg, P. H., Leader-Williams, N., Petracca, L., & Moilanen, A. (2016). Global priorities for national carnivore conservation under land use change. *Scientific Reports*, 6(April), 23814. <https://doi.org/10.1038/srep23814>
- Feranec, J., Hazeu, G., Christensen, S., & Jaffrain, G. (2007). Corine land cover change detection in Europe (case studies of the Netherlands and Slovakia). *Land Use Policy*, 24(1), 234–247. <https://doi.org/10.1016/j.landusepol.2006.02.002>
- Fick, S. E., & Hijmans, R. J. (2017). WorldClim 2: new 1-km spatial resolution climate surfaces for global land areas. *International Journal of Climatology*, 37(12), 4302–4315. <https://doi.org/10.1002/joc.5086>
- Goldman-Benner, R. L., Benitez, S., Boucher, T., Calvache, A., Daily, G., Kareiva, P., Kroeger, T., & Ramos, A. (2012). Water funds and payments for ecosystem services: practice learns from theory and theory can learn from practice. *Oryx*, 46(01), 55–63. <https://doi.org/10.1017/S0030605311001050>
- Grantham, H. S., Pressey, R. L., Wells, J. a, & Beattie, A. J. (2010). Effectiveness of biodiversity surrogates for conservation planning: different measures of effectiveness generate a kaleidoscope of variation. *PloS One*, 5(7), e11430. <https://doi.org/10.1371/journal.pone.0011430>
- Grima, N., Singh, S. J., Smetschka, B., & Ringhofer, L. (2016). Payment for Ecosystem Services (PES) in Latin America: Analysing the performance of 40 case studies. *Ecosystem Services*, 17, 24–32. <https://doi.org/10.1016/j.ecoser.2015.11.010>
- Guisan, A., Thuiller, W., & Zimmermann, N. E. (2017). Habitat Suitability and Distribution Models: With Applications in R. In *Ecology, Biodiversity and Conservation*. <https://doi.org/DOI:10.1017/9781139028271>
- Hadley, A. S., & Betts, M. G. (2016). Refocusing Habitat Fragmentation Research Using Lessons from the Last Decade. *Current Landscape Ecology Reports*, 1(2), 55–66. <https://doi.org/10.1007/s40823-016-0007-8>
- Hunter, L. (2011). *Carnivores of the world* (First edit). Princeton University Press.
- IDEAM, IGAC, IAvH, Invemar, I Sinchi, & IIAP. (2007). *Ecosistemas continentales, costeros y marinos de*

- Colombia* (First edit). Instituto de Hidrología, Meteorología y Estudios Ambientales, Instituto Geográfico Agustín Codazzi, Instituto de Investigación de Recursos Biológicos Alexander von Humboldt, Instituto de Investigaciones Ambientales del Pacífico Jhon von Neumann, Instituto.
- Inskip, C., & Zimmermann, A. (2009). Human-felid conflict: a review of patterns and priorities worldwide. *Oryx*, 43(01), 18. <https://doi.org/10.1017/S003060530899030X>
- Kramer-Schadt, S., Niedballa, J., Pilgrim, J. D., Schröder, B., Lindenborn, J., Reinfelder, V., Stillfried, M., Heckmann, I., Scharf, A. K., Augeri, D. M., Cheyne, S. M., Hearn, A. J., Ross, J., Macdonald, D. W., Mathai, J., Eaton, J., Marshall, A. J., Semadi, G., Rustam, R., ... Wilting, A. (2013). The importance of correcting for sampling bias in MaxEnt species distribution models. *Diversity and Distributions*, 19(11), 1366–1379. <https://doi.org/10.1111/ddi.12096>
- Le Saout, S., Hoffmann, M., Shi, Y., Hughes, A., Bernard, C., Brooks, T. M., Bertzky, B., Butchart, S. H. M., Stuart, S. N., Badman, T., Rodrigues, A. S. L., & Saout, S. Le. (2013). Conservation. Protected areas and effective biodiversity conservation. *Science (New York, N.Y.)*, 342(6160), 803–805. <https://doi.org/10.1126/science.1239268>
- Levin, N., Mazar, T., Brokovich, E., Jablon, P.-E., & Kark, S. (2010). Sensitivity analysis of conservation targets in systematic conservation planning. *Ecological Applications*, 25(7), 1997–2010. <https://doi.org/10.1890/14-1464.1>
- Liu, C., Berry, P. M., Dawson, T. P., & Pearson, R. G. (2005). Selecting thresholds of occurrence in the prediction of species distributions. *Ecography*, 28(2005), 385–393. <https://doi.org/10.1002/ece3.1878>
- MADS. (2014). *Guía Técnica para la Formulación de los Planes de Ordenación y Manejo de Cuencas Hidrográficas (POMCA)*. Dirección de Gestión Integral del Recurso Hídrico, Fondo Adaptación, Instituto de Hidrología, Meteorología y Estudios Ambientales - IDEAM - Subdirección de Estudios Ambientales.
- Mateo-Tomás, P., Olea, P. P., Sánchez-Barbudo, I. S., & Mateo, R. (2012). Alleviating human-wildlife conflicts: Identifying the causes and mapping the risk of illegal poisoning of wild fauna. *Journal of Applied Ecology*, 49(2), 376–385. <https://doi.org/10.1111/j.1365-2664.2012.02119.x>
- McRae, B. H., Dickson, B. G., Keitt, T. H., & Shah, V. B. (2008). Using circuit theory to model connectivity in ecology, evolution, and conservation. *Ecology*, 89(10), 2712–2724. <http://www.ncbi.nlm.nih.gov/pubmed/18959309>
- Miller, T. R., Minter, B. A., & Malan, L. C. (2011). The new conservation debate: The view from practical ethics. *Biological Conservation*, 144(3), 948–957. <https://doi.org/10.1016/j.biocon.2010.04.001>
- Moilanen, A., Wilson, K. A., & Possingham, H. P. (2009). Spatial conservation prioritization : quantitative methods and computational tools. In *Oxford biology*. Oxford University Press.
- Morales-Hidalgo, D., Oswalt, S. N., & Somanathan, E. (2015). Status and trends in global primary forest, protected areas, and areas designated for conservation of biodiversity from the Global Forest Resources Assessment 2015. *Forest Ecology and Management*, 352, 68–77. <https://doi.org/10.1016/j.foreco.2015.06.011>
- Moros, L., Corbera, E., Vélez, M. A., & Flechas, D. (2020). Pragmatic conservation: Discourses of payments for ecosystem services in Colombia. *Geoforum*, 108, 169–183. <https://doi.org/10.1016/j.geoforum.2019.09.004>
- Muscarella, R., Galante, P. J., Soley-Guardia, M., Boria, R. A., Kass, J. M., Uriarte, M., & Anderson, R. P. (2014). ENMeval: An R package for conducting spatially independent evaluations and estimating optimal model complexity for Maxent ecological niche models. *Methods in Ecology and Evolution*, 5(11), 1198–1205. <https://doi.org/10.1111/2041-210X.12261>
- Nagy-Reis, M., Oshima, J. E. de F., Kanda, C. Z., Palmeira, F. B. L., de Melo, F. R., Morato, R. G., Bonjorne, L., Magioli, M., Leuchtenberger, C., Rohe, F., Lemos, F. G., Martello, F., Alves-Eigenheer, M., da

- Silva, R. A., Silveira dos Santos, J., Priante, C. F., Bernardo, R., Rogeri, P., Assis, J. C., ... Ribeiro, M. C. (2020). NEOTROPICAL CARNIVORES: a data set on carnivore distribution in the Neotropics. *Ecology*, 101(11). <https://doi.org/10.1002/ecy.3128>
- Ocampo-Peñuela, N., & Pimm, S. L. (2014). Setting Practical Conservation Priorities for Birds in the Western Andes of Colombia. *Conservation Biology*, 28(5), 1260–1270. <https://doi.org/10.1111/cobi.12312>
- Peterson, A. T., Soberón, J., Pearson, R. G., Anderson, R. P., Martínez-Meyer, E., Nakamura, M., & Bastos Araujo, M. (2011). Ecological niches and geographic distributions. In *Choice Reviews Online* (Vol. 49, Issue 11). <https://doi.org/10.5860/CHOICE.49-6266>
- Phillips, S. J., Anderson, R. P., & Schapire, R. E. (2006). Maximum entropy modeling of species geographic distributions. *Ecological Modelling*, 190(3–4), 231–259. <https://doi.org/10.1016/j.ecolmodel.2005.03.026>
- Pitman, R. T., Swanepoel, L. H., Hunter, L., Slotow, R., & Balme, G. A. (2015). The importance of refugia, ecological traps and scale for large carnivore management. *Biodiversity and Conservation*, 24(8), 1975–1987. <https://doi.org/10.1007/s10531-015-0921-9>
- Radosavljevic, A., & Anderson, R. P. (2014). Making better Maxent models of species distributions: Complexity, overfitting and evaluation. *Journal of Biogeography*, 41(4), 629–643. <https://doi.org/10.1111/jbi.12227>
- Redacción Comercial del Espectador. (2017, December 11). La estrategia que le paga a los campesinos por cuidar los ecosistemas de Colombia. *El Espectador*. <https://www.elespectador.com/es-el-momento-de-los-que-transforman/noticias/medio-ambiente/la-estrategia-que-le-paga-los-campesinos-por-cuidar-los-ecosistemas-de-colombia-articulo-717493>
- Ripple, W. J., Chapron, G., López-Bao, J. V., Durant, S. M., Macdonald, D. W., Lindsey, P. A., Bennett, E. L., Beschta, R. L., Bruskotter, J. T., Campos-Arceiz, A., Corlett, R. T., Darimont, C. T., Dickman, A. J. M. Y. J., Dirzo, R., Dublin, H. T., Estes, J. A., Everatt, K. T., Galetti, M., Goswami, V. R., ... Wirsing, A. J. (2016). Saving the World ' s Terrestrial Megafauna. *BioScience*, XX(X), 1–6. <https://doi.org/10.1093/biosci/biw092>
- Ripple, W. J., Estes, J. a, Beschta, R. L., Wilmers, C. C., Ritchie, E. G., Hebblewhite, M., Berger, J., Elmhagen, B., Letnic, M., Nelson, M. P., Schmitz, O. J., Smith, D. W., Wallach, A. D., & Wirsing, A. J. (2014). Status and ecological effects of the world's largest carnivores. *Science*, 343(6167), 1241484–1–11. <https://doi.org/10.1126/science.1241484>
- Rodríguez, N., Armenteras, D., & Retana, J. (2015). Land Use Policy National ecosystems services priorities for planning carbon and water resource management in Colombia. *Land Use Policy*, 42, 609–618. <https://doi.org/10.1016/j.landusepol.2014.09.013>
- Rondinini, C., Wilson, K. a, Boitani, L., Grantham, H., & Possingham, H. P. (2006). Tradeoffs of different types of species occurrence data for use in systematic conservation planning. *Ecology Letters*, 9(10), 1136–1145. <https://doi.org/10.1111/j.1461-0248.2006.00970.x>
- Ruckelshaus, M., McKenzie, E., Tallis, H., Guerry, A., Daily, G., Kareiva, P., Polasky, S., Ricketts, T., Bhagabati, N., Wood, S. a., & Bernhardt, J. (2013). Notes from the field: Lessons learned from using ecosystem service approaches to inform real-world decisions. *Ecological Economics*. <https://doi.org/10.1016/j.ecolecon.2013.07.009>
- Salzman, J., Bennett, G., Carroll, N., Goldstein, A., & Jenkins, M. (2018). The global status and trends of Payments for Ecosystem Services. *Nature Sustainability*, 1(3), 136–144. <https://doi.org/10.1038/s41893-018-0033-0>
- Santamaría, M., Areiza, A., Matallana, C., Solano, C., & Galán, S. (2018). *Estrategias complementarias de conservación en Colombia* (1st ed.). Instituto Humboldt, Resnatur y Fundación Natura. http://oldpage.humboldt.gov.co/images/Cartilla_Conservacion_Aprobacion.pdf

- Santini, L., Saura, S., & Rondinini, C. (2016). A composite network approach for assessing multi-species connectivity: An application to road defragmentation prioritisation. *PLoS ONE*, *11*(10), 1–15. <https://doi.org/10.1371/journal.pone.0164794>
- Saura, S., Bodin, Ö., & Fortin, M.-J. (2014). Stepping stones are crucial for species' long-distance dispersal and range expansion through habitat networks. *Journal of Applied Ecology*, *51*(1), 171–182. <https://doi.org/10.1111/1365-2664.12179>
- Shcheglovitova, M., & Anderson, R. P. (2013). Estimating optimal complexity for ecological niche models: A jackknife approach for species with small sample sizes. *Ecological Modelling*, *269*, 9–17. <https://doi.org/10.1016/j.ecolmodel.2013.08.011>
- SINAP. (2019). *Hacia una política para el Sistema Nacional de Áreas Protegidas de Colombia Visión 2020-2030*.
- Soberón, J. (2007). Grinnellian and Eltonian niches and geographic distributions of species. *Ecology Letters*, *10*(12), 1115–1123. <https://doi.org/10.1111/j.1461-0248.2007.01107.x>
- Suárez-Castro, A. F., & Ramírez-Chaves, H. E. (2015). *Los carnívoros terrestres y semiacuáticos continentales de Colombia. Guía de Campo* (A. F. Suárez-Castro & H. E. Ramírez-Chaves (eds.); First edit, Issue November). Editorial Universidad Nacional de Colombia.
- Thornton, D., Zeller, K., Rondinini, C., Boitani, L., Crooks, K., Burdett, C., Rabinowitz, A., & Quigley, H. (2016). Assessing the umbrella value of a range-wide conservation network for Jaguars (*Panthera onca*). *Ecological Applications*, *26*(4), 1112–1124. <https://doi.org/10.5061/dryad.8507n>
- Venter, O., Hovani, L., Bode, M., & Possingham, H. (2013). Acting Optimally for Biodiversity in a World Obsessed with REDD+. *Conservation Letters*, *6*, 410–417. <https://doi.org/10.1111/conl.12018>
- Wallach, A. D., Bekoff, M., Nelson, M. P., & Ramp, D. (2015). Promoting predators and compassionate conservation. *Conservation Biology*, *29*(5), 1481–1484. <https://doi.org/10.1111/cobi.12525>
- Whitehead, A. L., Kujala, H., Ives, C. D., Gordon, A., Lentini, P. E., Wintle, B. a, Nicholson, E., & Raymond, C. M. (2014). Integrating Biological and Social Values When Prioritizing Places for Biodiversity Conservation. *Conservation Biology: The Journal of the Society for Conservation Biology*, *28*(4), 992–1003. <https://doi.org/10.1111/cobi.12257>
- Wilson, D. E., & Mittermier, R. A. (Eds.). (2009). *The handbook of mammals of the world. Vol1. Carnivores*. (First edit). Linx Edicions.
- Wilson, J. W., Bergl, R. A., Minter, L. J., Loomis, M. R., & Kendall, C. J. (2019). The African elephant *Loxodonta spp* conservation programmes of North Carolina Zoo: two decades of using emerging technologies to advance in situ conservation efforts. In *International Zoo Yearbook* (Vol. 53, Issue 1, pp. 151–160). Blackwell Publishing Ltd. <https://doi.org/10.1111/izy.12216>
- Wilson, K. a., Westphal, M. I., Possingham, H. P., & Elith, J. (2005). Sensitivity of conservation planning to different approaches to using predicted species distribution data. *Biological Conservation*, *122*(1), 99–112. <https://doi.org/10.1016/j.biocon.2004.07.004>
- Zárrate-Charry, D. A., Massey, A. L., González-Maya, J. F., & Betts, M. G. (2018a). Multi-criteria spatial identification of carnivore conservation areas under data scarcity and conflict: a jaguar case study in Sierra Nevada de Santa Marta, Colombia. *Biodiversity and Conservation*, *27*(13), 3373–3392. <https://doi.org/10.1007/s10531-018-1605-z>
- Zárrate-Charry, D. A., Massey, A. L., González-Maya, J. F., & Betts, M. G. (2018b). Multi-criteria spatial identification of carnivore conservation areas under data scarcity and conflict: a jaguar case study in Sierra Nevada de Santa Marta, Colombia. *Biodiversity and Conservation*. <https://doi.org/10.1007/s10531-018-1605-z>
- Zeller, K. A., McGarigal, K., & Whiteley, A. R. (2012). Estimating landscape resistance to movement: A review. *Landscape Ecology*, *27*(6), 777–797. <https://doi.org/10.1007/s10980-012-9737-0>

Appendix C

September 28, 2021
Dr. Jeremy Sanders
Editor-in-Chief

Andrew Dunn
Senior Publishing Editor

Dr Agnieszka Latawiec
Associate Editor

Pete Smith
Subject Editor

Anita Kristiansen
Editorial Coordinator
Royal Society Open Science

Please find attached/enclosed the revised version of our manuscript "*Connectivity conservation at the crossroads: protected areas vs payments for ecosystem services in conserving connectivity for Colombian carnivores, Colombia*" for publication in *Royal Society Open Science*. We thank you and the reviewers for the valuable comments. We have addressed the reviewer commentary and have made two new reviews by English speaking academic partners to ensure that the document present a clear narrative

We really appreciate your willingness to consider this revised version. We have included a point-by-point response to the reviewers' comments below. Color codes correspond to individual changes made to the manuscript to address the reviewer suggestions when it was possible. Most of the changes are based on narrative and comments of English-speaking academic partners, these are maintained in Track Changes for your consideration.

Thank you very much once again for this opportunity, and we look forward to your final decision.

Sincerely,

Diego Zárrate-Charry, Ph.D.
Research coordinator ProCAT-Colombia/Sierra to Sea Institute
Researcher Forest Biodiversity Research Network
Cellphone Colombia: (+57) 3177992028
dazarrate@procat-conservation.org, godiezcharry@gmail.com
www.procat-conservation.org

The comments made by the editor or reviewers are in italics and our answers in red font. If our answer resulted in changes to the manuscript, we include the text in quotations.

Response to Reviewer number 1:

Reviewer: 1

I believe the authors have adequately addressed all the major comments raised by me and the other reviewer in the previous round. I have a few minor comments that authors need to make before this paper can be accepted for publication.

Abstract:

L27-28: This line says 16 of 22 carnivores but elsewhere in text the authors say 16 of 24. Please correct as necessary.

Thank you very much, there is an error in the second number, currently there are 22 reported carnivores for the Colombian Caribbean region. We include here the list just for reference. We made the change in the text in the (page 3 Line 26)

Family	Species
Ursidae	Tremarctos ornatus
Canidae	Speothos venaticus
Canidae	Urocyon cinereoargenteus
Canidae	Cerdocyon thous
Felidae	Leopardus tigrinus
Felidae	Herpailurus yagouaroundi
Felidae	Leopardus weidii
Felidae	Leopardus pardalis
Felidae	Puma concolor
Felidae	Panthera onca
Mephitidae	Conepatus semistriatus
Mustelidae	Mustela frenata
Mustelidae	Lontra longicaudis
Mustelidae	Eira barbara
Mustelidae	Galictis vittata
Procyonidae	Procyon cancrivorus
Procyonidae	Potos flavus
Procyonidae	Nasua nasua
Procyonidae	Nasuella olivacea
Procyonidae	Bassaricyon medius
Procyonidae	Bassaricyon alleni
Procyonidae	Bassaricyon neblina

Introduction:

In trying to incorporate reviewer comments, I think the flow/narrative of the text has become a bit distorted. The authors have done a good job in including information that was missing from the previous version. But I suggest reworking the text in this section to make it more streamlined.

Thanks a lot for the comment, indeed it was challenging to make the proposed changes and maintain a cohesive narrative. The text was reviewed by two exterior native English-speaking reviewers and several changes were made. We hope such changes improve the way the message is presented.

For example, Page 2 L17-24 should come after Page 2 L51. Similarly, I think the authors should try and structure the paragraphs such that they are thematically linear (each para should address 1-2 key themes; start with broader picture and narrow down to Colombia and carnivores).

We moved the paragraph to ensure coherence within the introduction regarding theme and geographic scale. We also made several changes in the document to ensure a better understanding after the review.

Methods:

Again, please split paragraphs carefully such that each para has 1-2 central themes with relevant citations.

We made the same process of review within this paragraph to ensure better understanding.

Page 6 L18: "Aichi"

We corrected the word.

Results:

All subheadings are labeled "4.1". Please correct these.

We corrected the subheadings numerations of the results section.

Page 8 L10, L20, L46: It would be better to start these paragraphs by adding one line that reiterates what aspect was being examined and how. Example, "Based on potential distribution of species mapped using climatic variables under the MaxEnt framework, we found...";

As proposed, we made the changes in the start of the paragraphs. We include here the relation where we made the changes. (Page 8 line 34, Page 8 line 45, page 8 line 52)

Page 9 L24: I suggest providing a more informative sub-heading.

We included a more detailed sub-heading and apply it both in the methods and in the result section.

Discussion:

Same comment here as with the Introduction. Please restructure/reorganize the text so that the narrative is coherent and linear. Start with the key insights from your analysis, move on to highlight species-specific

insights, then discuss the broader issues with PAs versus PES schemes in Colombia, draw parallels to research from other landscapes or locations, add 1-2 paragraphs of limitations, and finally a short, may be 1 or 2 paragraphs of conclusions.

We made changes to the text and tried to include and change phrasing to ensure a better reading after a review by native speaking researchers. We also changed the way part of the conclusions were written to ensure a better reading.

Figure 5: These results are interesting and should be discussed in some more detail (see my comment above on species-specific insights in the Discussion).

We included a couple of sentences regarding species specific results, but without make in it too long based on our idea to show general results for the group and avoid species specific results based on previous comments from reviewers. The inclusion can be found in the page 11 line 38

Finally, I still found a lot of typographical and grammatical errors throughout the manuscript (missing periods, commas or other punctuation marks, spelling errors, sentences starting with a small letter, etc.). Some sentences are also framed awkwardly and need to be rewritten. I understand that the MS was already proofread by someone who was not involved in the study, but I encourage the authors to make edits as suggested above and seek assistance/guidance with language once again.

We have addressed the reviewer commentary and have made two new reviews by English speaking academic partners to ensure that the document avoid such errors.